# Advancing Mathematical Reasoning in Language Models: The Impact of Problem-Solving Data, Data Synthesis Methods, and Training Stages

**Zui Chen**[1,3,*] **Tianqiao Liu**[2,*] **Mi Tian**[2], **Qing Tong**[2], **Weiqi Luo**[1], **Zitao Liu**[1,†]
[1]Jinan University [2]TAL Education Group [3]ShanghaiTech University
{chenzui3, liutianqiao1, tianmi, tongqing}@tal.com
{liuzitao, lwq}@jnu.edu.cn

## Abstract

Mathematical reasoning remains a challenging area for large language models (LLMs), prompting the development of math-specific LLMs such as LLEMMA, DeepSeekMath, and Qwen2-Math, among others. These models typically follow a two-stage training paradigm: pre-training with math-related corpora and post-training with problem datasets for supervised fine-tuning (SFT). Despite these efforts, the improvements in mathematical reasoning achieved through continued pre-training (CPT) are often less significant compared to those obtained via SFT. This study addresses this discrepancy by exploring alternative strategies during the pre-training phase, focusing on the use of problem-solving data over general mathematical corpora. We investigate three primary research questions: (1) Can problem-solving data enhance the model's mathematical reasoning capabilities more effectively than general mathematical corpora during CPT? (2) Are synthetic data from the same source equally effective, and which synthesis methods are most efficient? (3) How do the capabilities developed from the same problem-solving data differ between the CPT and SFT stages, and what factors contribute to these differences? Our findings indicate that problem-solving data significantly enhances the model's mathematical capabilities compared to general mathematical corpora. We also identify effective data synthesis methods, demonstrating that the tutorship amplification synthesis method achieves the best performance. Furthermore, while SFT facilitates instruction-following abilities, it underperforms compared to CPT with the same data, which can be partially attributed to its poor learning capacity for more challenging problem-solving data. These insights provide valuable guidance for optimizing the mathematical reasoning capabilities of LLMs, culminating in our development of a powerful mathematical base model called MathGPT-8B[1].

## 1 Introduction

To address the challenge of insufficient mathematical reasoning capabilities in large language models (LLMs), various math-specific LLMs are developed. These include models that enhance performance from the pre-training stage, such as LLEMMA (Azerbayev et al., 2023), DeepSeekMath (Shao et al., 2024), InternLM-Math (Ying et al., 2024), and Qwen2-Math (Yang et al., 2024a), as well as models that improve through post-training, such as MetaMath (Yu et al., 2023), WizardMath (Luo et al., 2023), and KwaiYiiMath (Fu et al., 2023). These models generally follow a common training paradigm. During the pre-training stage, math-related corpora are filtered from extensive internet data to augment the model's mathematical knowledge. During the post-training stage, they

---

*These authors contributed equally. Work done during Zui Chen's internship at Guangdong Institute of Smart Education, Jinan University, Guangzhou, China.

†The corresponding author.

[1]Model is available at https://huggingface.co/ai4ed/MathGPT-8B

typically utilize problem datasets and their augmented versions, such as Program-of-Thought (PoT) (Chen et al., 2022), evol-Instruct (Xu et al., 2023), and Tool-Integrated Reasoning (TIR) (Gou et al., 2023; Yin et al., 2024), to construct supervised datasets for Supervised Fine-Tuning (SFT). This enables the models to follow instructions and produce outputs in the desired format. Recently, there is a growing focus on constructing preference datasets for the solution process to perform Step-DPO (Lai et al., 2024) or online-RLHF (Dong et al., 2024). These approaches aim to obtain more accurate reasoning pathways, thereby significantly enhancing the mathematical reasoning capabilities of the models.

Due to the intrinsic distinction between mathematical knowledge and general world knowledge, different strategies are required for their effective acquisition and application. The primary challenge in acquiring world knowledge lies in memorizing and understanding vast amounts of information, necessitating large corpora during the pre-training phase to enhance knowledge reserves (Roberts et al., 2020; Petroni et al., 2019; Dubey et al., 2024). In contrast, mathematical knowledge involves a relatively limited set of elements, concepts, axioms, and theorems that need to be memorized and understood. The real challenge often lies not in recalling the relevant knowledge but in using this knowledge for reasoning or planning (Hao et al., 2023).

From previous studies, it might seem that the continued pre-training (CPT) stage contributes less to mathematical reasoning abilities. However, recent studies, such as Physics of LLM (Allen-Zhu & Li, 2023) and MiniCPM (Hu et al., 2024), highlight the importance of teaching models how to utilize memorized knowledge during the pre-training stage. These findings question the effectiveness of the prevalent paradigm for enhancing mathematical reasoning abilities, which primarily focuses on memorizing more mathematical knowledge during the pre-training phase and developing reasoning abilities in the post-training phase. Therefore, we propose that alternative strategies utilizing mathematical problems and their reasoning steps—referred to as problem-solving data—during the pre-training phase, to teach the model how to apply its memorized knowledge rather than simply increasing the volume of relevant data, could potentially lead to significant improvements in mathematical reasoning capabilities. With these considerations, we aim to explore the following fundamental research questions (RQs):

**RQ1**: During the CPT stage, can providing problem-solving data more effectively enhance the model's mathematical reasoning capabilities compared to using general mathematical corpora?

**RQ2**: If problem-solving data can enhance mathematical reasoning capabilities, are synthetic data from the same source equally effective, and what synthesis methods are most efficient?

**RQ3**: How do the capabilities developed from the same problem-solving data differ between the CPT and SFT stages, and what factors contribute to these differences?

We address these three research questions separately. In Section 3, we explore RQ1 by comparing the impact of using problem-solving data and examining various math data mixture ratios, which leads to Result 1. In Section 4, we investigate RQ2 by delving into four data synthesis techniques: response diversification, query expansion, retrospective enhancement, and tutorship amplification, resulting in Result 2. In Section 5.1, we address RQ3 by first identifying, from a holistic perspective, the differences in learning mathematical capabilities between the CPT and SFT stages using problem-solving data. Subsequently, in Section 5.2 and Section 5.3, we further analyze RQ3 by dividing the problem-solving data into subsets based on data distribution and difficulty level to investigate the sources of these differences, ultimately leading to Results 3-5.

**Result 1**: Providing math problem-solving data significantly enhances the model's mathematical capabilities compared to general mathematical corpora and a higher proportion of problem-solving data is more effective.

**Result 2**: Response diversification, query expansion, and tutorship amplification were effective. Among these, tutorship amplification methods emerged as distinctly superior, leveraging a teacher model to identify and correct errors based on the student model's responses, aiming to equip the model with self-correction capabilities.

**Result 3**: Overall, while SFT can facilitate some learning of mathematical capabilities, it has a clear disadvantage compared to CPT.

**Result 4**: From the perspective of data distribution, both SFT and CPT primarily develop capabilities aligned with their data distributions. However, SFT's in-domain (IND) learning ability is weaker than that of CPT. Regarding out-of-domain (OOD) capability learning, the conclusions are less clear, with only the observation that SFT is more susceptible to disturbances from data distribution compared to CPT.

**Result 5**: From the perspective of difficulty level, providing more challenging problem-solving data enables more effective learning, with this advantage being particularly evident in CPT compared to SFT. This may be the primary source of the learning capability differences between CPT and SFT. Therefore, we recommend preparing more challenging problem-solving data for the CPT phase.

After addressing our three RQs, we identify the optimal strategy combination and apply it to the Llama3-8B model (Dubey et al., 2024), resulting in the highly efficient MathGPT-8B. MathGPT-8B surpasses various math-specific models including DeepSeekMath-Base-7B (Shao et al., 2024) and Qwen2-Math-7B (Yang et al., 2024a), and exhibits capabilities comparable to Qwen2-Math-72B and the recently released Qwen2.5-Math-7B (Yang et al., 2024b). We introduce only 100B mathematical tokens, equivalent to 1/10 of Qwen2.5-Math-7B, and perform CPT based on a weaker base model. This validates that our proposed method is a more efficient approach for enhancing mathematical capabilities compared to existing paradigms. Additionally, MathGPT-8B retains strong general knowledge capabilities, as confirmed by MMLU (Hendrycks et al., 2020) benchmarks. Since no post-training is conducted, we are releasing the base version of MathGPT-8B, allowing the research community to perform further post-training to enhance its capabilities.

## 2 EXPERIMENTAL PREPARATION

In this section, we provide a comprehensive overview of experimental preparations, including data, baseline models, and metrics.

**Training Data.** The training data is categorized into three groups: (1) **General corpus**, which includes scientific texts from the ArXiv subset of RedPajama (Weber et al., 2023), code datasets from AlgebraicStack (Azerbayev et al., 2023) and StarCoder (Li et al., 2023), and natural language datasets from the C4 and Wikipedia subsets of RedPajama (Weber et al., 2023), to prevent catastrophic forgetting and maintain robustness. (2) **Mathematical corpus**, which utilizes corpus on mathematical content like OpenWebMath (Paster et al., 2023) to improve mathematical proficiency. (3) **Problem-solving data**, which includes NuminaMath (Li et al., 2024), Lila (Mishra et al., 2022), and proprietary data, with 14 million pieces used for synthetic data augmentation. Our experiments employ 48.3B tokens from the general corpus, 13.7B from the mathematical corpus, 7.2B from problem-solving data, and 30.54B from synthetic data. Detailed descriptions are provided in Appendix A.1.

**Base Model.** We select Llama2 (Touvron et al., 2023) as our base model to ensure robustness in our findings, as it predates the release of OpenWebMath (Paster et al., 2023). By choosing a model that existed before the introduction of recent mathematical corpora, we effectively mitigate the risk of contamination from these newer datasets. More details are provided in Appendix A.2.

**Evaluation Set.** To minimize contamination of the data set and expand the capacity assessment, we expand our evaluation set to include GAOKAO and ZHONGKAO, along with GSM8K (Cobbe et al., 2021) and MATH (Hendrycks et al., 2021). GAOKAO and ZHONGKAO datasets, developed after the release of Llama2, enable the measurement of a wider range of abilities. Detailed descriptions of the data sets are provided in Appendix A.3.

**Deduplication and Decontamination.** We use the MinHash deduplication (Lee et al., 2022) framework to enhance training data quality by removing documents with significant duplicate content. This process includes setting specific byte thresholds for deduplication and decontamination, effectively eliminating contaminated documents, particularly from OpenWebMath (Paster et al., 2023). More details are provided in Appendix A.4.

**Evaluation Metrics.** Our evaluation follows a three-stage process: model inference using zero-shot and few-shot prompts, answer comparison to handle irregular outputs, and statistical scoring to determine accuracy. In the statistical scoring stage, we select the higher accuracy between the zero-shot and few-shot approaches for each dataset to ensure the reliability and robustness of the

results, given that some models perform better in zero-shot settings while others prefer few-shot settings. We report the arithmetic mean of accuracy scores across datasets. Detailed methodologies are discussed in Appendix A.5.

# 3 IMPROVING REASONING ABILITY IN CPT WITH PROBLEM-SOLVING DATA

We believe that, compared to simply remembering and understanding more mathematical knowledge from vast corpora, the focus of mathematical knowledge acquisition during the pre-training phase primarily lies in learning to apply this knowledge for reasoning or planning. The intuitive approach is to provide corresponding data for problem-solving. Therefore, in this section, we first aim to validate RQ1, specifically the effectiveness of providing problem-solving data during the CPT phase. This serves not only as a validation of our main argument but also as the foundation for subsequent research questions. We then continue to explore the impact of the proportion of problem-solving data to determine an appropriate data ratio and verify the efficiency of providing problem-solving data.

**Experiments.** We design four experimental groups, including one base group and three test groups. Our goal is to demonstrate the effectiveness of providing problem-solving data by comparing the base group with the test groups, while exploring suitable data mixing ratios through comparisons among the three test groups. Specifically, the total amount of math data used in the base group and test groups remains the same, with the base group utilizing the math corpus as its math data. In contrast, the test groups employ a mix of the math corpus and problem-solving data as their math data, with the mixing ratios varied among the three test groups. The specific data details are as follows: where the **data mixture ratio** indicates the mixing proportion of general data to math data, and the **math data mixture ratio** reflects the blending proportion of the math corpus to problem-solving data. More experimental design discussions can be found in the Appendix C.

- **Base1**: Using 48.3B general corpus and 14.7B math corpus, mixed in a 4:6 ratio.
- **Test1**: Using 48.3B general corpus, 7.5B math corpus, and 7.2B problem-solving data, with data mixture ratio 4:6, math data mixture ratio 5:5.
- **Test2**: Same as Test1, but using a math data mixture ratio of 3:7.
- **Test3**: Same as Test1, but using a math data mixture ratio of 7:3.

**Training Details.** We utilize Llama2 (Touvron et al., 2023) as the base model and perform CPT for 25,000 steps, with a global batch size of 1024 and a context length of 4096 tokens. The learning rate is warmed up to 1e-4 and then decays to 1e-5 using a cosine schedule (Loshchilov & Hutter, 2022). The training data is split into 95% for training and 5% for validation. After completing the 25,000 steps, we select the checkpoint with the lowest validation loss for evaluation as the result.

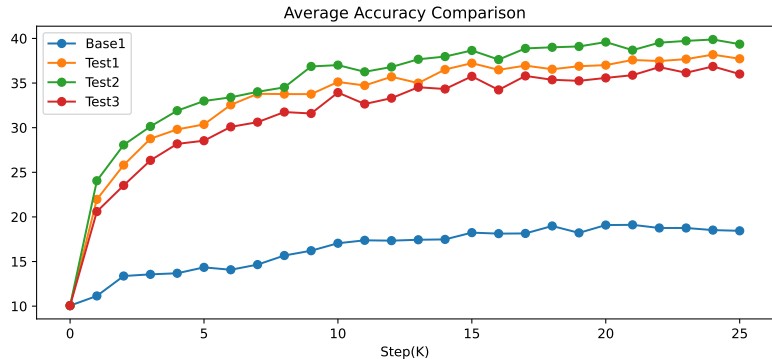

Figure 1: The average accuracy of the four groups varies with the number of steps.

**Results.** As shown in Figure 1, the blue line, representing the reference group following the current training paradigm, indicates that CPT using the math corpus effectively improves problem-solving accuracy. However, compared to the other three curves, even though Base1 utilizes the same number of tokens, the trend and extent of improvement in mathematical capabilities are significantly lower

than those of the three test groups. For the three test groups, the green line in Figure 1 shows that as the number of steps increases, its average accuracy consistently surpasses the other two. Notably, we do not introduce new tokens but simply alter the math data mixture ratio. In Appendix D, we include an additional test group, Test4, which uses only problem-solving data. Although it does not utilize any math corpus tokens, it achieves performance comparable to or even higher than that of the green line. Additionally, we report the accuracy of four evaluation sets for further comparison. Thus, we achieve **Result 1: Providing math problem-solving data significantly enhances the model's mathematical capabilities compared to general mathematical corpora, and a higher proportion of problem-solving data is more effective.**

## 4 EXPLORATION OF EFFICIENT DATA SYNTHESIS METHODS

In the preceding sections, Results 1 highlights the effectiveness of problem-solving data. However, the limited availability of such data compared to internet data underscores the need for efficient data synthesis methods. Additionally, it is not yet fully researched whether further synthesis from the same problem-solving data during the pre-training stage can enhance model performance. To address these issues and RQ2, we explore four data synthesis methods: response diversification, query expansion, retrospective enhancement, and tutorship amplification. Our aim is to validate the effectiveness of synthesized data and identify the most efficient synthesis method. Below, we briefly introduce the data synthesis methods used in our study.

**Response Diversification** aims to enhance model capabilities by generating diverse reasoning paths through methods like rejection sampling. Since it does not alter the answers, response diversification does not require additional labeling, making it easy to implement. The effectiveness of response data synthesis is established through various implementations (Yuan et al., 2023; Yu et al., 2023; Chen et al., 2024). Instead of using a sampling-then-deduplication approach, we require the model to follow two steps to improve the efficiency of response diversification: (1) Generate two distinct solutions based on the question and the original answer; (2) Select the solution with the correct final answer to serve as one diversified training sample.

**Query Expansion** aims to enhance model capabilities by expanding the question set. However, generating high-quality questions directly is challenging. Existing methods (e.g., Yu et al., 2023 and Mitra et al., 2024) leverage the concept of reshaping, which involves generating new questions based on existing questions and answers through rephrasing, reversing statements, and other techniques. The synthesis of new questions focuses on ensuring: (1) the accuracy of the newly generated questions, and (2) the accuracy of their corresponding answers. We integrate existing methods and emphasize these key points by requiring the LLM to perform augmentation in four steps based on the input question and solution: (1) transform the question into a statement, (2) generate new questions based on the statement, (3) provide answers for the new questions, and (4) evaluate the answers and explain the reasoning. Our approach improves quality through three main aspects: first, we provide the original questions and answers; second, steps 1 and 2 ensure that the generated questions are valid and solvable; and third, steps 3 and 4 involve self-evaluation to assess the quality of the answers to the new questions.

**Retrospective Enhancement** Ye et al. (2024) posits that teaching the model to directly correct mistakes is beneficial. They employ a low-resource construction method that involves directly inserting subsequent steps into preceding ones, allowing models to retry upon regret. A special [back] token is used for identification, which is why we refer to it as retrospective enhancement. This method is validated on GSM8K using a small parameter model with minimal pre-training. Our scenario differs in two key ways: (1) we utilize a more diverse question set, with some questions significantly different from the simpler forms in GSM8K; (2) we perform CPT on a mainstream model that possesses a certain level of mathematical capability. We aim to validate the effectiveness of this straightforward method.

**Tutorship Amplification** is inspired by the real-life practice of teachers guiding students to rectify mistakes. As evidenced by OpenAI (2024), models can be trained to spot errors. This agrees with Ye et al. (2024), who suggest that while models can detect errors, they lack opportunities for correction. Tutorship amplification simulates a realistic error correction process. In this process, a "strong" model, acting as a teacher, aids a "weak" model, representing a student. After the student model generates an answer to a problem, the teacher model performs the following actions: it checks

whether the student's answer is correct. If the answer is correct, it responds affirmatively. Otherwise, it points out the erroneous steps and continues solving from that point. We aim for this process to achieve three objectives: first, to construct realistic errors that are likely to occur; second, to enable self-evaluation and error identification; third, to facilitate timely correction of identified mistakes. We believe these three elements will aid the model in learning self-correction and enhancing its reasoning accuracy.

**Synthetic Data.** A seed set is created by filtering subsets from the original problem-solving data, based on the completeness of data and the number of reasoning steps involved. Following this, four data synthesis methods are applied to the seed set. Details regarding the quantity of the resulting synthetic data and associated token counts are provided in Table 1.

**Experiment.** We utilize a control group, Base2, which comprises 48.3B general corpus tokens, 14.7B math corpus tokens, and 7.2B problem-solving data. In addition to the data used in Base2, we introduce extra tokens generated from the four data synthesis methods to establish four experimental groups. These models are continuous pre-trained from the raw Llama2 base model. Each data combination is trained for at most 25,000 steps, and the checkpoint at which the validation set loss converged is selected. The final accuracy is then evaluated based on this chosen checkpoint. Other training parameters are consistent with those in Section 3.

| Model | Num | Tokens | GSM8K | MATH | GAOKAO | ZHONGKAO | Average |
|---|---|---|---|---|---|---|---|
| Base2 | - | - | 47.84 | 20.12 | 22.98 | 67.05 | 39.50 |
| Res-Div | 14,018,544 | 6.82B | 52.99 | 23.22 | 23.83 | 65.15 | 41.30 |
| Query-Exp | 24,459,192 | 4.78B | 51.25 | 23.08 | 27.23 | 69.13 | 42.67 |
| Retro-Enh | 14,707,792 | 5.04B | 45.11 | 21.72 | 22.98 | 66.67 | 39.12 |
| Tutor-Amp | 11,942,328 | 13.90B | 64.44 | 35.88 | 32.77 | 69.32 | 50.60 |

Table 1: Performance comparison of four experimental groups using different synthetic data methods and one control group across four evaluation sets. "Num" denotes the count of problem-solving questions and corresponding solutions used, while "Tokens" indicates the total number of tokens. The model abbreviations represent: Res-Div (Response Diversification), Query-Exp (Query Expansion), Retro-Enh (Retrospective Enhancement), and Tutor-Amp (Tutorship Amplification).

**Results.** The experimental results for the four combinations of synthetic data are presented in Table 1. From this, we derive **Result 2: Response Diversification, Query Expansion and Tutorship Amplification emerge as effective data synthesis techniques, with Tutorship Amplification registering particularly pronounced effects**. Conversely, Retrospective Enhancement appears to exert minimal influence. We postulate that this could be attributed to the fact that the erroneous data constructed is not grounded in actual sampling, resulting in a lower likelihood of occurrence and thereby inhibiting the model's capacity for error detection and rectification learning. We also notice that query expansion and response diversification yield limited enhancements. We propose one hypothesis for this observation: during data generation, the model's self-evaluation might have failed to identify its own errors, thereby constraining the quality of the synthesized data. As for the effectiveness of Tutorship Amplification, our hypotheses are twofold: first, the model acquires a reasoning framework for self-checking, error detection, and correction through the tutorship amplification data; second, the tutorship amplification data facilitates the learning of knowledge application to correctly resolve problems via error correction.

## 5 ABILITIES ACQUISITION COMPARISON OF CPT AND SFT STAGES

In the previous two sections, we have demonstrated that providing problem-solving data during the CPT phase efficiently teaches the model to apply mathematical knowledge and enhances its reasoning ability. However, how does this differ from developing mathematical reasoning skills during the SFT phase? In this section, we first verify that the change in the training stage indeed raises the upper limits of the model's capability, not merely due to the data. Then, we investigate the sources of differences in mathematical learning between the CPT and SFT phases from two perspectives: data distributions and difficulty levels.

## 5.1 Comparison of Abilities Acquisition

In this section, we explore how the stage at which problem-solving data is used (CPT vs. SFT) significantly affects the model's ultimate capabilities. We have a total of 7.2B problem-solving data, which can be allocated at either the CPT or SFT stage. Additionally, we sample 0.072B problem-solving data for 1%-SFT to endow the model with instruction-following ability. We propose the following experimental settings to compare the acquisition of learning capabilities between the CPT and SFT stages:

- **Base1**: CPT with 48.3B general corpus and 14.7B math corpus.
- **Base2**: CPT with 48.3B general corpus, 7.5B math corpus, and 7.2B problem-solving data.
- **Base1-SFT**: SFT with 7.2B problem-solving data based on Base1.
- **Base1-1%SFT**: SFT with 0.072B problem-solving data based on Base1.
- **Base2-1%SFT**: SFT with 0.072B problem-solving data based on Base2.

It is important to note that we perform SFT on both Base1 and Base2 using 1% of the problem-solving data. This setup allows us to isolate the impact of instruction-following capability improvements and thereby assess the true enhancement in mathematical reasoning ability brought about by introducing problem-solving data at the CPT stage.

**Experiment Details.** During the SFT stage, we set a batch size of 256 and a learning rate that decayed from 1e-5 to 1e-6 following a cosine schedule. We train for 3 epochs, ensuring that the training loss converged. After convergence, we select the optimal result from 10 checkpoints for reporting, which typically occurred around the checkpoints at 2 epochs. More experimental design discussions can be found in the Appendix C.

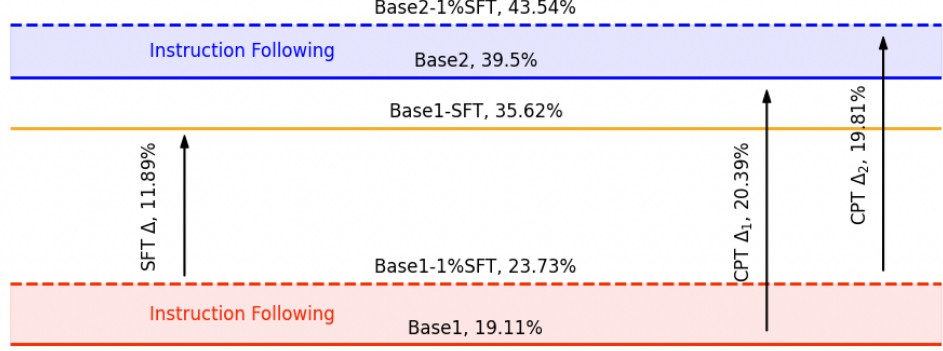

Figure 2: Comparison of the acquisition of learning capabilities between the CPT and SFT stages

**Results.** The evaluation results across the four datasets can be found in Appendix E. Their average accuracy is illustrated in Figure 2. First, we observe the red and blue shaded areas, where a small amount of SFT data brought similar improvements on both Base1 and Base2. From the evaluation results, this improvement stems from a significant reduction in the model's previously inconsistent and repetitive outputs. We believe this is a result of the supervised approach in SFT, leading to leading to an interesting conclusion: **A small amount of SFT data is sufficient to enhance the model's ability to follow instructions**.

Next, we compare the results after removing the influence of instruction-following capabilities. At this point, the differences, denoted as SFT $\Delta$ and CPT $\Delta_2$, can be viewed as the improvements in mathematical reasoning ability obtain during the SFT and CPT phases, respectively. Given that both use the same data, but the capability gain in SFT is only about 60% of that achieves during CPT. Additionally, comparing Base1-SFT and Base2, despite using the same data, Base1-SFT also gains the ability to follow instructions, yet its performance is still inferior to Base2. Thus we conclude **Result 3: Overall, while SFT can facilitate some learning of mathematical capabilities, it has a clear disadvantage compared to CPT**.

To better understand SFT's impact on learning capabilities, we add three additional experimental groups, where we perform SFT with 10%, 20%, and 50% splits of the problem-solving data. We compare these with Base1, 1% SFT, and 100% SFT to analyze the effect of SFT data volume on

reasoning improvement. The results are shown in Figure 6. We observe a significant increase in average accuracy at the 1% SFT markgroup, followed by a logarithmic-linear relationship between data volume and accuracy improvement. This further validates that a small amount of SFT data enhances the model's ability to follow instructions. Moreover, increasing the SFT data may continue to logarithmically improve the model's reasoning ability.

## 5.2 IMPACT OF DIFFERENT DATA DISTRIBUTIONS

In the previous section, we observe that the reasoning capability learned during the SFT phase is significantly weaker compared to CPT. In this section, we aim to explore the source of this difference. Our intuition is that data distributions might have different impacts on capability learning at each stage, with CPT possibly contributing to enhanced out-of-distribution (OOD) performance. However, our findings contradict this hypothesis. Both CPT and SFT primarily develop capabilities aligned with the data distributions they are trained on.

**Experiment.** We design our experiments by segmenting the training data based on evaluation sets. Specifically, we select one evaluation set to represent in-distribution (IND) capabilities, with the remaining sets are considered out-of-distribution (OOD). Correspondingly, we retain only the portions of the training data aligned with IND capabilities. However, it is important to note two key challenges: first, during the decontamination process, we already exclude any data that overlapped with the evaluation sets; second, the scope of mathematical abilities inherently includes overlap and coverage across different areas. Due to these factors, it is challenging to perfectly match training data to specific capabilities. Therefore, we utilize knowledge point labels from the original problem-solving data to segment out 0.83B middle school data, corresponding to ZHONGKAO as its IND capabilities, and 0.89B high school data, corresponding to GAOKAO as its IND capabilities. The OOD capabilities are represented by the remaining evaluation sets that do not align with these IND capabilities. More experimental design discussions can be found in the Appendix C. The specific experimental design is as follows:

- **Base1**: As described in Section 3. CPT with 48.3B general corpus and 14.7B math corpus.
- **Middle-school-SFT**: SFT with 0.83B middle school data on Base1.
- **Middle-school-CPT**: CPT with Base1 data and middle school data.
- **High-school-SFT**: SFT with 0.89B high school data on Base1
- **High-school-CPT**: CPT with Base1 data and high school data.

| Model | GSM8K | MATH | GAOKAO | ZHONGKAO | Average |
|---|---|---|---|---|---|
| Base1 | 28.20 | 9.48 | 8.09 | 30.68 | 19.11 |
| Middle-school-SFT | 22.67 (-5.53) | 16.36 (+6.88) | 10.21 (+2.12) | **52.28 (+21.60)** | 25.38 (+6.27) |
| Middle-school-CPT | 29.42 (+1.22) | 15.04 (+5.56) | 8.09 (0.00) | **54.71 (+24.03)** | 26.81 (+7.70) |
| High-school-SFT | 19.11 (-9.09) | 13.48 (+4.00) | **16.60 (+8.51)** | 36.78 (+6.10) | 21.49 (+2.38) |
| High-school-CPT | 23.96 (-4.24) | 13.82 (+4.34) | **22.98 (+14.89)** | 34.19 (+3.51) | 23.74 (+4.63) |

Table 2: Learning capabilities analysis across various data distributions.

**Results.** As shown in Table 2, for the IND capabilities represented by **bolded evaluation results**, learning during the CPT stage consistently leads to greater improvements compared to learning during the SFT stage. This effect is especially evident in the learning of more challenging high school-level knowledge. In addition, for OOD capabilities, learning during the SFT stage experiences significantly more disruption. This is particularly noticeable for GSM8K (see the data distribution and capability dimension chart in Appendix B), which has the largest distributional difference. After SFT, the model's performance on OOD tasks suffers more compared to CPT. Thus, we achieve **Result 4: Both SFT and CPT primarily develop capabilities aligned with their data distributions. However, SFT's IND learning ability is weaker than that of CPT. Regarding OOD capability learning, the conclusions are less clear, with only the observation that SFT is more susceptible to disturbances from data distribution compared to CPT.**

## 5.3 IMPACT OF DIFFERENT DIFFICULTY LEVELS

In the previous section, although we clarify that both CPT and SFT involve in-domain capability learning, it remains unclear what cause SFT's learning performance to be weaker than CPT's. However, conclusions in Result 4 are more evident in the high school training data compared to middle school, prompting us to explore the difference in learning capabilities between CPT and SFT with varying difficulty levels problem-solving data.

**Experiment.** We select a 5B subset of our problem-solving data and categorize it based on the number of solution reasoning steps: data requiring 1-3 steps is classified as easy, 4-7 steps as medium, and 8 or more steps as hard. The distribution of samples account for 36.0%, 38.4%, and 25.6% of the total data, respectively, while token counts make up 23.0%, 36.0%, and 41.0%, respectively. Given the unavoidable inaccuracies in this method of categorization, we focus solely on easy data and hard data for the CPT and SFT comparison experiments. More experimental design discussions can be found in the Appendix C. The experimental groups are designed as follows:

- **Base1**: As described in Section 3. CPT with 48.3B general corpus and 14.7B math corpus.
- **Easy-SFT**: SFT using the easy data subset on top of Base1.
- **Easy-CPT**: CPT incorporating both the Base1 data and the easy data subset.
- **Hard-SFT**: SFT using the hard data subset on top of Base1.
- **Hard-CPT**: CPT incorporating both the Base1 data and the hard data subset.

| Model | GSM8K | MATH | GAOKAO | ZHONGKAO | Average | Easy | Medium | Hard |
|---|---|---|---|---|---|---|---|---|
| Base1 | 28.20 | 9.48 | 8.09 | 30.68 | 19.11 | 14.86 | 6.69 | 4.85 |
| Easy-SFT | 31.31 | 14.46 | 14.04 | 48.30 | 27.03 | 22.52 (+7.66) | 10.68 (+4.00) | 6.94 (+2.09) |
| Easy-CPT | 37.98 | 15.70 | 17.02 | 52.46 | 30.79 | 27.61 (+12.75) | 13.33 (+6.64) | 6.27 (+1.42) |
| Hard-SFT | 31.39 | 17.40 | 15.32 | 54.55 | 29.66 | 24.37 (+9.51) | 11.93 (+5.24) | 6.84 (+1.99) |
| Hard-CPT | 45.79 | 23.96 | 26.38 | 69.89 | 41.51 | 35.78 (+20.92) | 20.17 (+13.48) | 9.32 (+4.47) |

Table 3: Performance comparison of CPT and SFT models on different difficulty levels. The table shows the average and specific performance on easy, medium, and hard data subsets.

**Results.** The results in the left half of Table 3, which is divided by vertical lines, show that CPT models consistently outperform SFT models, with some relative improvements specifically indicated. Notably, Hard-CPT exhibits greater relative enhancements compared to Easy-CPT, and these improvements are not limited to just the hard domain accuracy but are observed across all datasets. Moreover, regardless of whether it is SFT or CPT, training on hard data consistently yields better results compared to training on easy data. This suggests **Result 5: Providing more challenging problem-solving data enables more effective learning, and this advantage is particularly evident in CPT compared to SFT. This may be the primary source of the learning capability differences between CPT and SFT. Therefore, given limited computational resources, we recommend preparing more challenging problem-solving data for the CPT phase.**

The results in right half of Table 3 indicate that both SFT and CPT models achieve their highest improvements on Easy problems, with reduced gains as problem difficulty increases. For example, Easy-SFT and Easy-CPT show significant improvements of +7.66 and +12.75 on Easy problems, but only +2.09 and +1.42 on hard problems, respectively. Similarly, Hard-SFT and Hard-CPT exhibit their largest gains on easy problems (+9.51 and +20.92) compared to hard problems (+1.99 and +4.47). These patterns suggest that **Regardless of the training data's difficulty, both SFT and CPT primarily focus on learning to solve simpler, fewer-step problems**.

## 6 TRAINING A STRONG MATH-SPECIFIC MODEL

To further validate the effectiveness of our empirical results, we aim to train a strong math-specific model based on the Llama3-8B (Dubey et al., 2024), named **MathGPT-8B**. We follow the conclusions from the three RQs outlined earlier: (1) We maintain a 3:7 ratio of mathematical corpus to problem-solving data; (2) We use synthesized data from Query Expansion, Response Diversification, and Tutorship Amplification, with a focus on expanding data using the most efficient Tutorship Amplification method; (3) We filter and expand the raw data by focusing on problems with more than

five reasoning steps, using these as seed data to generate additional synthesized data. In addition, we incorporate newly released mathematical corpora (Han et al., 2024) into the training. Ultimately, we use 39.6B general corpus tokens, 46.7B mathematical corpus tokens, and 51.1B problem-solving data and synthesized data tokens to train **MathGPT-8B** for 25,000 steps, with a global batch size of 1024 and a context length of 8192 tokens. The learning rate is warmed up to 1e-4 and then decayed to 1e-5 using a cosine schedule.

**Results.** As presented in Table 4, compared to the base model, we significantly enhance the foundational capabilities of Llama3-8B, even surpassing larger models such as Llama3.1-70B and Qwen2-72B, which have over 70 billion parameters. Additionally, we evaluate our model using the Gao et al. (2024) on the MMLU (Hendrycks et al., 2020) benchmarks, achieving a score of 0.6222 compared to Llama3-8B's 0.6211, demonstrating that it maintained its general knowledge capabilities.

Compared to math-specific base models, MathGPT-8B outperforms DeepSeekMath-Base-7B (Shao et al., 2024) and Qwen2-Math-7B (Yang et al., 2024a), and exhibits capabilities comparable to Qwen2-Math-72B and the recently released Qwen2.5-Math-7B (Yang et al., 2024b). Compared to Qwen2.5-Math-7B, MathGPT-8B is trained on only 140 billion tokens (100 billion of which are math-related), while Qwen2.5-Math-7B utilizes 1 trillion tokens, as reported. Additionally, MathGPT-8B starts from a weaker base model. These findings validate our proposed method as an efficient approach to enhancing mathematical capabilities compared to existing paradigms. Further discussions on related work can be found in Appendix F. Since we do not perform a complete post-training process, we are releasing the base version of our model. This allows the research community to conduct further post-training to enhance its capabilities as needed.

| Model | GSM8K | MATH | GAOKAO | ZHONGKAO | Average |
|---|---|---|---|---|---|
| **General Models** | | | | | |
| Llama3-8B | 58.38 | 17.04 | 13.62 | 42.61 | 32.91 |
| Llama3-70B | 82.34 | 38.42 | 28.09 | 64.02 | 53.21 |
| Llama3.1-8B | 56.79 | 19.70 | 11.49 | 44.70 | 33.17 |
| Llama3.1-70B | 81.73 | 39.66 | 31.06 | 64.77 | 54.31 |
| Qwen2-7B | 80.44 | 47.82 | 27.23 | 70.45 | 56.49 |
| Qwen2-72B | 86.58 | 56.88 | 45.11 | 73.67 | 65.56 |
| Qwen2.5-7B | 84.61 | 53.22 | 45.53 | 80.30 | 65.92 |
| Qwen2.5-72B | 90.60 | 59.38 | 56.60 | 82.95 | 72.38 |
| **Math-specific Models** | | | | | |
| LLEMMA-7B | 41.47 | 18.94 | 14.89 | 45.08 | 30.10 |
| DeepSeekMath-Base-7B | 65.73 | 33.40 | 23.83 | 62.69 | 46.41 |
| Qwen2-Math-7B | 80.67 | 53.02 | 42.13 | 77.08 | 63.22 |
| Qwen2-Math-72B | 88.63 | 61.88 | 51.91 | 81.25 | 70.92 |
| Qwen2.5-Math-7B | 85.44 | 59.10 | 53.19 | 78.79 | 69.13 |
| Qwen2.5-Math-72B | 88.70 | 67.10 | 53.62 | 81.63 | 72.76 |
| **(MathGPT-8B)** | **81.20** | **60.38** | **60.43** | **80.49** | **70.62** |

Table 4: Model performance metrics (General and math-specific models)

# 7 CONCLUSION

In this study, we investigate the enhancement of mathematical reasoning capabilities in LLMs through alternative pre-training strategies. Our findings lead to the development of MathGPT-8B, a competitive model that outperforms most 7B models and exhibits capabilities comparable to much larger models despite being trained on fewer tokens. Future work should expand in two key areas. First, we need to refine the data synthesis methods. Although we have demonstrated the effectiveness of synthetic data, our current approaches are relatively naive. Second, we should explore the role and impact of alignment processes during post-training. Investigating these aspects will help further improve the mathematical reasoning capabilities of the model.

## ACKNOWLEDGMENTS

This work was supported in part by National Key R&D Program of China, under Grant No. 2022YFC3303600; in part by NFSC under Grant No. 62477025; in part by Key Laboratory of Smart Education of Guangdong Higher Education Institutes, Jinan University (2022LSYS003) and in part by Beijing Municipal Science and Technology Project under Grant No. Z241100001324011.

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

## APPENDIX

## A   DETAILED EXPERIMENT PREPARATION

### A.1   TRAINING DATA DETAILS

The training data utilize in our study is categorized into three distinct groups: (1) **General corpus**, encompassing scientific texts from the ArXiv subset of RedPajama (Weber et al., 2023), code datasets from AlgebraicStack (Azerbayev et al., 2023) and StarCoder (Li et al., 2023), along with natural language datasets from the C4 and Wikipedia subsets of RedPajama (Weber et al., 2023). The inclusion of general data helps prevent the model from experiencing catastrophic forgetting, where it might lose previously acquired knowledge during specialized training. Moreover, maintaining a broad base of general knowledge ensures the stability and robustness of the model, enabling it to retain a well-rounded understanding and perform effectively across various tasks. (2) **Mathematical corpus** is designed to enhance the model's proficiency in mathematics, primarily comprising

general mathematical content extracted from sources like CommonCrawl web pages. The main objective is to imbue the pre-trained model with foundational mathematical knowledge, including terminology, theorems, proofs, etc. To achieve this, we utilize OpenWebMath (Paster et al., 2023), a resource shown to effectively improve mathematical capabilities, as demonstrated in (Azerbayev et al., 2023). (3) **Problem-solving data**, which we believe can more efficiently enhance the model's reasoning abilities. We collect 25 million pieces of problem-solving data, including those from open-source resources such as NuminaMath (Li et al., 2024) and Lila (Mishra et al., 2022), as well as proprietary data. Among them, 14 million pieces are used as seed data for augmentation to create our synthetic data. Overall, using the Llama2 (Touvron et al., 2023) to conduct experiments on RQs, we employ 48.3B tokens from the general corpus, 13.7B from the mathematical corpus, 7.2B from problem-solving data and 30.54B from synthetic data.

## A.2 BASE MODEL SELECTION

The selection of the base model is pivotal in shaping our conclusions, as it directly influences the reliability and applicability of our findings. To ensure that our exploration of research questions yields practically valuable insights, we have chosen to base our study on mainstream models. Considering that OpenWebMath may have been widely incorporated into recent LLMs, introducing this mathematical corpus might not produce the desired effect. Therefore, we select Llama2 (Touvron et al., 2023), which is released prior to OpenWebMath (Paster et al., 2023), as our base model. This decision aims to enhance the robustness of our conclusions.

## A.3 EVALUATION DATASETS

Considering both the risk of dataset contamination and the scope of capabilities, we expand the evaluation set to include GAOKAO and ZHONGKAO, in addition to GSM8K (Cobbe et al., 2021) and MATH (Hendrycks et al., 2021). The GAOKAO dataset comprises both GAOKAO-2023 and GAOKAO-2024, derived from the most recent Chinese National College Entrance Examinations. We convert the problem format into math word problems, translate the questions, and retain 235 items after review. Similarly, the ZHONGKAO dataset is sourced from the 2023 Chinese High-School Entrance Examination and includes 658 translated math word problems. Both GAOKAO and ZHONGKAO datasets are created after the release of Llama2 (Touvron et al., 2023), which strengthens our conclusion. These additional datasets provide coverage of different dimensions of ability compared to GSM8K and MATH. From the perspectives of general knowledge, math knowledge, and reasoning steps. GAOKAO is similar to MATH but demands more general knowledge, while ZHONGKAO is akin to GSM8K but may require more mathematical knowledge and fewer reasoning steps. Detailed ability dimensions can be found in Appendix B. We believe this expanded evaluation set will lead to a more comprehensive assessment and serve as a valuable reference for subsequent improvement.

## A.4 DEDUPLICATION AND DECONTAMINATION

We use the MinHash deduplication Lee et al. (2022) framework to remove entire documents containing duplicate text that exceeds a certain threshold from the training data. Specifically, we set a threshold of 2048 bytes for deduplication to improve the quality of the training data. Additionally, we set a threshold of 100 bytes to remove any data from the training set that contains more than 100 bytes of overlapping text with subsets of the train and test sets in the evaluation data. We believe this can account for some contamination caused by simple paraphrasing. (Notably, in the case of OpenWebMath (Paster et al., 2023), we remove 2594 contaminated documents, which have a significant impact on the conclusions during our initial experiments.)

## A.5 EVALUATION METRICS

The evaluation process comprises three stages: model inference, answer comparison, and statistical scoring. During model inference, we utilize both zero-shot and few-shot prompt templates for each dataset. For the zero-shot approach, we employ a simple Chain-of-Thought (CoT) prompt (Kojima et al., 2022). In the few-shot approach, we use 8-shot and 4-shot settings for the GSM8K and MATH datasets, respectively, and apply the same few-shot settings from GSM8K and MATH to

the ZHONGKAO and GAOKAO datasets. For answer comparison, we use an answer comparison model [2] to address issues related to the irregular output of the base models, such as inconsistent stopping criteria and extracting answers from CoT prompts. In the statistical scoring stage, we select the higher accuracy between the zero-shot and few-shot approaches for each dataset to ensure the reliability and robustness of the results, given that some models perform better in zero-shot settings while others prefer few-shot settings. Finally, we report the arithmetic mean of the accuracy scores across the four datasets as the average accuracy.

## B  ABILITY DIMENSIONS OF THE FOUR EVALUATION SETS

GSM8K, MATH, ZHONGKAO, and GAOKAO, four evaluation sets, are introduced to enrich the dimensions of the evaluation, as shown in Table 5 with example problems.

To preliminarily understand the differences in capabilities across various dimensions of the evaluation process, we attempt to define three capability dimensions: general knowledge, math knowledge, and reasoning steps. As Table 6 illustrates, each capability dimension is divided into three levels, with requirements progressively increasing from Level 1 to Level 3. General knowledge describes the demands for understanding common sense, such as the fact that a day consists of 24 hours; math knowledge refers to the complexity of mathematical knowledge, including arithmetic, elementary, and advanced mathematics; reasoning steps describe the depth of reasoning. Figure 3 displays the performance of the four evaluation sets across different dimensions. Overall, GAOKAO and MATH represent similar capability dimensions, but GAOKAO might require some general knowledge for certain problems. ZHONGKAO and GSM8K both demand a higher level of general knowledge, but differ in their requirements for math knowledge and reasoning steps.

Furthermore, as shown in Figure 4, we analyze the data distribution of problems in the datasets to clarify the data distribution of different evaluation sets and the impact of different data distributions on out-of-distribution (OOD) capabilities as discussed in Section 5.2. Specifically, we sample up to 1,000 problems from the evaluation sets and used t-SNE for dimensionality reduction, with the visualization shown in 4(a) and the cosine similarity situation in 4(b). It is evident that MATH, ZHONGKAO, and GAOKAO have certain correlations, whereas GSM8K exhibits the largest distributional difference. This may also explain why different evaluation sets perform differently in terms of OOD capabilities, as discussed in Table 2 and related conclusions.

| Dataset | Problem |
|---------|---------|
| GSM8K | Josh decides to try flipping a house. He buys a house for \$80,000 and then puts in \$50,000 in repairs. This increased the value of the house by 150%. How much profit did he make? |
| MATH | How many vertical asymptotes does the graph of $y = \frac{2}{x^2+x-6}$ have? |
| ZHONGKAO | What is the opposite number of $4$? |
| GAOKAO | Given the sets $M = \{x|x + 2 >= 0\}$, $N = \{x|x - 1 < 0\}$, what is $M \cap N =$? |

Table 5: Example problems from four evaluation sets

## C  DISCUSSION ON EXPERIMENTAL SETTINGS

Our experimental design generally adheres to the principles of comparative experiments, forming control groups to test hypotheses by introducing variations. Below, we elaborate on the considerations behind the design of each experimental group.

**In Section 3**, to mitigate the influence of the total amount of math data used on the experimental conclusions, as described in the main text, we control the total amount of math data used in both the base group and the test groups to be the same. Specifically, the base group uses the entire 14.7B math

---

[2]https://huggingface.co/Tianqiao/DeepSeek-7B-Math-Compare-Answer

| Competency Dimension | Level | Definition |
|---|---|---|
| General Knowledge | 1 | Involves minimal General Knowledge |
|  | 2 | Less than 50% of the problems require General Knowledge |
|  | 3 | More than 50% of the problems require General Knowledge |
| Math Knowledge | 1 | Basic arithmetic operations |
|  | 2 | Requirements for the Chinese High School Entrance Examination |
|  | 3 | Requirements for the Chinese National College Entrance Examinations |
| Reasoning Steps | 1 | Within 1-3 steps |
|  | 2 | Within 3-5 steps |
|  | 3 | More than 5 steps |

Table 6: Definitions of Competencies Across Different Levels

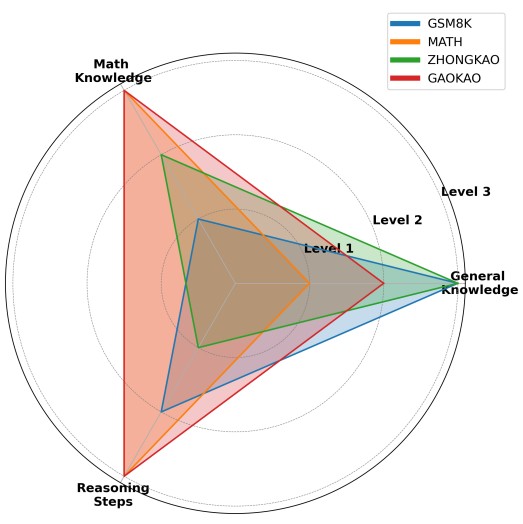

Figure 3: Ability dimensions of four evaluation sets

corpus as the math data, while the test groups use 7.2B problem-solving data and split the remaining 7.5B math corpus to ensure that the total math token was also 14.7B. Subsequently, in three test groups, we adjust the math data mixture ratio to further verify the effectiveness of problem-solving data and determine the optimal mixture ratio for subsequent experiments. Specifically, based on the token ratio of 7.5B:7.2B, we start with a math data mixture ratio of 5:5. Then, we adjust the ratio to 3:7, where the smaller 7.5B math corpus could be fully utilized within 10K steps and used more than twice within 25K steps. We believe this ensures the full utilization of data. Finally, we complement the experiment with a reverse ratio of 7:3.

**In Section 4**, to delineate the impact of different synthetic data, we introduce a control group, Base2, which used the entire problem-solving data on top of Base1. The experimental group further incorporate synthetic data into this setup. We aim to verify that the synthetic data contributed new value rather than merely substituting the original data.

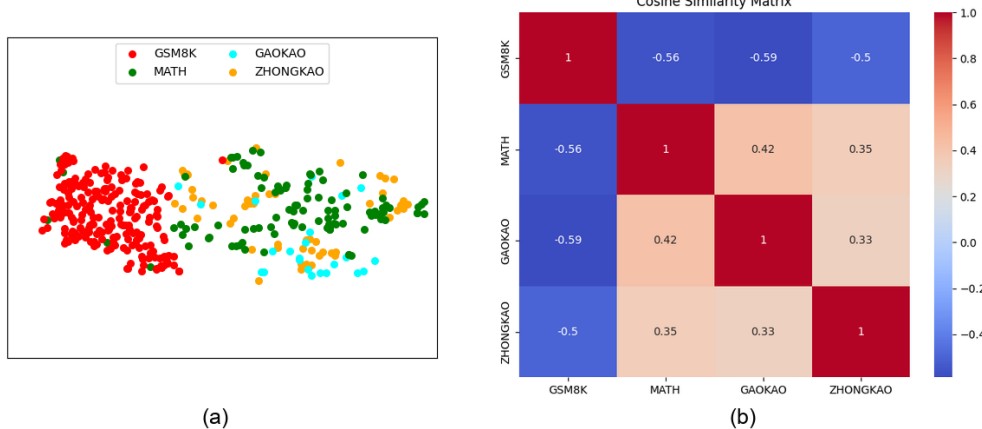

Figure 4: (a) Data distribution of problems of the four evaluation sets. (b) Dataset similarity based on data distribution calculation.

**In Section 5.1**, we compare how the stage at which problem-solving data is introduced (CPT vs. SFT) significantly affects the model's ultimate capabilities. First, we follow the setups of Base1 and Base2 and conduct SFT using the same data on Base1 to create a comparative experiment Base1-SFT. We hypothesize that Base1-SFT would benefit from enhanced instruction-following ability, which Base2 might lack. To validate this, we partition 1% of the data, assuming it has limited impact on reasoning ability but contributes to instruction-following ability. Subsequently, we apply this 1% data for SFT on both Base1 and Base2 groups. By comparing Base1-1%SFT with Base1-SFT, we evaluate the reasoning ability gained from SFT, and by comparing Base1-1%SFT with Base2-1%SFT, we assess the reasoning ability gained from CPT.

**In Section 5.2**, we primarily focus on differences in capabilities across the same evaluation dataset representing the data distribution at various training stages. To this end, Base1 is reused to define the improvement in the experimental groups' abilities and then we introduce two experimental groups, Middle-school-SFT and Middle-school-CPT, which use a Middle-school data subset from the training set for SFT and CPT, respectively, forming a comparison to evaluate the IND learning differences between SFT and CPT on a specific evaluation dataset. Additionally, the differences in OOD learning on other evaluation datasets are also analyzed. Subsequently, we replace the Middle-school subset with a High-school subset in the training set, implementing two additional experimental groups and repeating the same experiments to strengthen the robustness of the conclusions.

**In Section 5.3**, the experimental design is similar to that in Section 5.2, with a key difference: we focus on variations in learning ability of training data with different difficulty levels at different training stages. Thus, besides comparing SFT and CPT using easy and hard subsets of the training data separately, we also contrast the performance of different training subsets within the same training stage. Comparisons based on evaluation dataset distributions are used only as supplementary analysis.

# D    DETAILED RESULTS OF PROBLEM-SOLVING DATA EFFECTIVENESS EXPERIMENT

We hypothesize that simply performing CPT with the problem-solving data could be sufficient, and add an experimental group, Test4, which uses only problem-solving data as the math data, specifically:

• **Test4**: Using 48.3B general corpus and 7.2B problem-solving data, with data mixture ratio 4:6.

The results are shown in the Figure 5. We find that even with a smaller amount of math data, the purple line corresponding to Test4 and the green line corresponding to Test2, which represent the experimental groups using the smallest math data mixture, demonstrate consistent or even superior

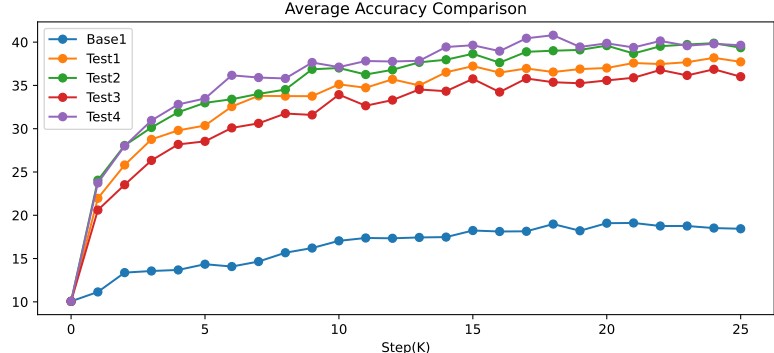

Figure 5: The average accuracy of the five groups varies with the number of steps.

| Model | GSM8K | MATH | GAOKAO | ZHONGKAO | Average |
|---|---|---|---|---|---|
| Llama2-7b | 14.40 | 5.10 | 4.26 | 16.48 | 10.06 |
| Base1 | 28.20 | 9.48 | 8.09 | 30.68 | 19.11 |
| Test1 | 44.88 | 19.72 | 20.00 | 66.29 | 37.72 |
| Test2 | 48.29 | 20.78 | 23.40 | 67.05 | 39.88 |
| Test3 | 42.15 | 19.48 | 22.55 | 63.26 | 36.86 |
| Test4 | 40.11 | 25.12 | 29.79 | 68.18 | 40.80 |

Table 7: Accuracy of the all experimental groups across the four evaluation set.

performance. This strongly complements Result 1, highlighting the effectiveness of problem-solving data, which can even surpass the impact of adding a large number of new tokens.

The detailed results of Base1, Test1, Test2, Test3 and Test4 are in Table 7. Notably, when the math corpus is not used, the specific metrics of Test4 across the four evaluation datasets no longer align with those of Test1-3. This shift in data distribution undermines the improvement in GSM8K performance while enhancing the improvements in MATH and GAOKAO capabilities.

## E    DETAILED RESULTS OF COMPARISON OF ABILITIES ACQUISITION

The evaluation results across the four datasets are in Table 8. And the relationship between average accuracy and SFT data quantity is in Figure 6.

## F    RELATED WORK

We discuss the related work on math CPT. LLEMMA (Azerbayev et al., 2023) initially focused on continuing pre-training to enhance mathematical reasoning capabilities, collecting open-source data including from OpenWebMath (Paster et al., 2023) and providing the Proof-Pile-2 dataset. They made preliminary attempts at continuous pre-training in the mathematics domain and shared their experiences. DeepSeekMath (Shao et al., 2024) advanced the effects of mathematical continuing pre-training by improving data quality, primarily training a fastText model to recall more OpenWebMath-like mathematical web pages and iterating this process, which also provided reliable experience for research beyond mathematical reasoning. InternLM-Math (Ying et al., 2024) utilized open-source datasets and internal datasets and trained a scoring model to identify high-quality datasets. Qwen2-Math (Yang et al., 2024a) and the more recent Qwen2.5-Math (Yang et al., 2024b) have begun to focus on using synthetic data, effectively achieving significant improvements.

| Model | GSM8K | MATH | GAOKAO | ZHONGKAO | Average |
|---|---|---|---|---|---|
| Base1 | 28.20 | 9.48 | 8.09 | 30.68 | 19.11 |
| Base1-1%SFT | 31.08 | 12.10 | 12.34 | 39.39 | 23.73 |
| Base1-10%SFT | 32.37 | 13.74 | 11.49 | 42.42 | 25.01 |
| Base1-20%SFT | 34.65 | 16.26 | 13.62 | 46.40 | 27.73 |
| Base1-50%SFT | 36.92 | 19.34 | 14.04 | 57.20 | 31.88 |
| Base1-SFT | 42.84 | 21.88 | 18.30 | 59.47 | 35.62 |
| Base2 | 47.84 | 20.12 | 22.98 | 67.05 | 39.50 |
| Base2-1%SFT | 51.40 | 27.10 | 25.96 | 69.70 | 43.54 |

Table 8: Model performance metrics with SFT

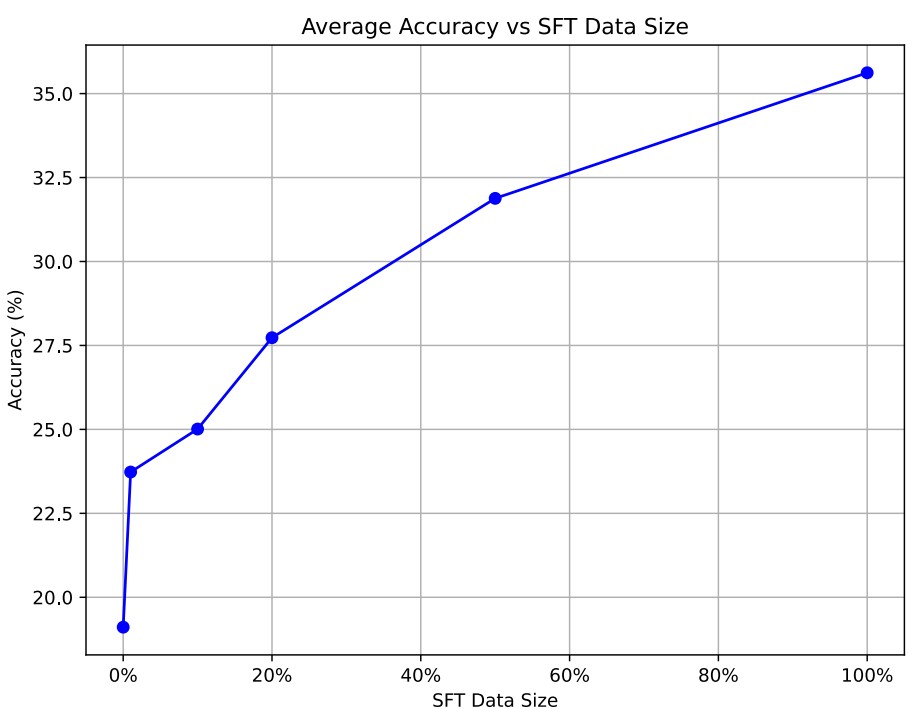

Figure 6: The relationship between average accuracy and SFT data quantity.

## G    SYNTHETIC DATA PROMPT

---

**Response Diversification**

You are a math teacher, please complete the following task using English.

**Your task is**: for the math problem below, in addition to the given solution, provide two more different solutions.

If you can provide two different solutions, start with $<response>$accept$</response>$, then present the additional solutions beginning with 'Solution2:' and 'Solution3:' respectively.

If you believe you cannot offer two different solutions or the solutions provided might be inaccurate, start with $<response>$refuse$</response>$.

**Please ensure the solutions are correct and distinct.** If you doubt the correctness of your solutions, or if it seems the problem does not allow for multiple solutions, directly indicate refusal by starting with $<response>$refuse$</response>$, and then explain the reason.

---

**Query Expansion**

Your goal is to create different math word questions and their solutions from a given question and its solution. You should follow these steps:

1. Convert the question into a statement and fill in $<statement>$ FILL IN HERE $</statement>$.
2. Create a new question based on the statement and fill in $<question>$ FILL IN HERE $</question>$.
3. Provide a solution to the new question and fill in $<solution>$ FILL IN HERE $</solution>$.
4. Check the solution and report $<check>$Accept$</check>$ or $<check>$Refuse$</check>$. And then fill the reason in $<reason>$ FILL IN HERE $</reason>$.
5. Repeat the process for a total of 2 questions and solutions.

---

**Tutorship Amplification**

As a mathematics teacher, please check the solution to the following math problem.

If the solution is correct, please only reply $<check>$correct$</check>$.

If the solution is incorrect, please first respond with $<check>$wrong$</check>$, then identify the erroneous steps, correct the errors, and continue to provide the correct solution.

Please note that your response must include $<check>$correct$</check>$ or $<check>$wrong$</check>$ at the beginning of the response.

---

