# OpenReview forum: "Advancing Mathematical Reasoning in Language Models: The Impact of Problem-Solving Data, Data Synthesis Methods, and Training Stages"
_ICLR.cc/2025/Conference — ICLR 2025 Poster_

### Official Review · Reviewer_XMnz · 2024-10-20

**Soundness:** 2
**Presentation:** 3
**Contribution:** 2
**Rating:** 6
**Confidence:** 5

**Summary:**

The paper titled **"Advancing Mathematical Reasoning in Language Models: The Impact of Problem-Solving Data, Data Synthesis Methods, and Training Stages"** explores the limitations of current pre-training strategies for mathematical reasoning in large language models (LLMs). It addresses three primary research questions related to improving the model’s mathematical reasoning capabilities: (1) Whether problem-solving data during the pre-training phase is more effective than general mathematical corpora, (2) The efficacy of synthetic data, and (3) How these strategies differ between continued pre-training (CPT) and supervised fine-tuning (SFT). The authors present a variety of data synthesis methods and analyze their impact on model performance. The results suggest that problem-solving data significantly enhances performance, particularly when coupled with effective data synthesis techniques like tutorship amplification, leading to the development of a competitive model, JiuZhang-8B.

**Strengths:**

- The paper introduces new insights into the importance of problem-solving data and explores under-researched data synthesis methods like tutorship amplification. It delves into an interesting research question regarding the effect of data mix on performance. Additionally, the study examines the differing impacts of continual pretraining and supervised fine-tuning on final mathematical performance.

  These findings are likely to be of  importance for further research in the fields of pretraining and supervised fine-tuning. The paper's exploration of these topics contributes valuable knowledge to the ongoing development of machine learning models and their application in problem-solving tasks.

- The experiments are thorough, with detailed comparisons between different data types, synthesis methods, and training stages.  Although the technical depth is substantial, the paper is mostly well-written, with a clear exposition of the research questions and a logical progression through the results.

- This work could have a  impact on the development of LLMs for complex reasoning tasks, especially in mathematics. The introduction of JiuZhang-8B as a model trained on fewer tokens is interesting.

**Weaknesses:**

- **Limited Practical Examples**: The paper could be enhanced by including more detailed, practical examples of how problem-solving data improves model performance. Concrete examples from the dataset would clarify the real-world impact of the proposed techniques.

- **Underexplored Larger Model Performance**:  The paper's experimental focus on mid-sized models like JiuZhang-8B, while informative, leaves a significant gap in understanding how the proposed methods perform and scale with larger language models. This limitation potentially understates the full impact and scalability of the presented approach.

  Expanding the analysis to include larger models in the 13B, 32B, or even 70B parameter range would provide crucial insights into the scalability and effectiveness of the proposed techniques. Such an expansion could reveal whether the benefits observed in mid-sized models persist, amplify, or perhaps even diminish when applied to more powerful language models.

  Moreover, the inclusion of larger models in the study might necessitate an increase in the volume of training data. This presents an opportunity to further explore the interplay between data quality and quantity in the context of more capacious models. It could potentially reveal new insights into the optimal balance between these factors for different model sizes, providing valuable guidance for future research。



- **Generalization to Other Domains**: The focus is heavily on mathematical reasoning, but there is little discussion on how the proposed techniques might generalize to other domains that also require reasoning.

- **Data Synthesis Techniques**: Although the tutorship amplification method was shown to be most effective, the explanations for why certain methods (e.g., retrospective enhancement) performed poorly could be expanded. Further exploration of why specific synthesis methods yielded limited results would strengthen the paper.

**Questions:**

- Could the authors provide more specific examples of problem-solving data used during pre-training and demonstrate how it differs from general mathematical corpora?
- Could the authors provide more insights into the limitations of the retrospective enhancement technique? What additional methods or adjustments could make this approach more effective?
-  Have the authors considered applying the problem-solving data approach to other reasoning-heavy domains, such as physics or logic-based tasks? How would the methods transfer to these fields?
- How do the proposed data synthesis methods, such as tutorship amplification, perform when applied to larger LLMs like 72B? Is the impact equally significant?
- Given the success of the tutorship amplification method, do the authors plan to further refine this approach? Could it be applied in real-time tutoring systems or adaptive learning models for broader applications?

---

> ### Author Response · Authors · 2024-11-24
>
> **Q1:** Could the authors provide more specific examples of problem-solving data used during pre-training and demonstrate how it differs from general mathematical corpora?
>
> **A1**: General mathematical corpora include a large volume of text on concepts, axioms, theorems, and other descriptive content, typically derived from filtered internet corpora. Examples include open-web-math [1] and InfiMM-WebMath [2]. In contrast, problem-solving data refers to datasets composed of mathematical problems and their reasoning processes. These datasets often involve the application of concepts, axioms, and theorems. An example is provided below:
>
> ---
>
> Question: In the numbers -1, +7, 0, -\frac{2}{3}, \frac{5}{16}, how many are positive numbers?
> A. 1
> B. 2
> C. 3
> D. 4
>
> Solution: Among the numbers -1, +7, 0, -\frac{2}{3}, \frac{5}{16}, there are +7 and \frac{5}{16}, which are positive numbers, totaling 2. Therefore, the correct answer is: B.
>
> ---
>
> [1] https://huggingface.co/datasets/open-web-math/open-web-math
>
> [2] https://huggingface.co/datasets/Infi-MM/InfiMM-WebMath-40B
>
>
>
> ### Q2: Could the authors provide more insights into the limitations of the retrospective enhancement technique? What additional methods or adjustments could make this approach more effective?
>
> **A2:** Firstly, we believe that the intuitive hypothesis behind the retrospective enhancement method—"allowing models to retry upon regret"—is quite reasonable. This is also why we conducted experiments and modified the process to develop the tutorship amplification method. We think its limitations may stem from the data construction process, where subsequent steps are directly inserted into previous steps to simulate "allowing models to retry upon regret." Although this indeed saves resources, the constructed data does not reflect actual errors, leading the model to learn how to correct non-existent mistakes. To address this, we attempted to construct some real errors, which led to the subsequently proposed tutorship amplification method.
>
> ### Q3: Have the authors considered applying the problem-solving data approach to other reasoning-heavy domains, such as physics or logic-based tasks? How would the methods transfer to these fields?
>
> **A3:** Similar to Reviewer 3zrA's Q1, we have conducted a detailed discussion; please refer to that.
>
> ### Q4: How do the proposed data synthesis methods, such as tutorship amplification, perform when applied to larger LLMs like 72B? Is the impact equally significant?
>
> **A4:** We have not conducted CPT on larger models. Due to the computational cost of training a 72B model, we do not plan to carry out comparative experiments. Before training the 72B model, we plan to prioritize improving the quality of the synthetic data methods and explore more synthesis methods.
>
> ### Q5: Given the success of the tutorship amplification method, do the authors plan to further refine this approach? Could it be applied in real-time tutoring systems or adaptive learning models for broader applications?
>
> **A5:** We plan to continue improving the tutorship amplification method but do not have plans to apply it in real-time tutoring systems. We believe the potential areas for improvement in tutorship amplification include:
>
> 1. **Enhancing the quality of synthetic data:** This includes filtering out instances where the student model's answer is incorrectly judged as incorrect and optimizing the teacher model's response quality by introducing methods like process supervision.
>
> 2. **Optimizing the formatting of synthetic data:** This involves adjusting prompts and post-processing steps to clearly define the erroneous parts, analysis parts, and problem-solving parts in the synthetic data. This would facilitate ablation experiments and simplify subsequent applications of format shaping, process supervision, and other reasoning techniques.

---

> > ### Comment · Reviewer_XMnz · 2024-11-29
> >
> > Thank you for your detailed responses to the questions. While the clarifications are helpful, my following concerns remain inadequately addressed:
> >
> > 1. Although specific problem-solving examples were shared, the broader integration of such examples to illustrate real-world impact is insufficient. A more comprehensive set of practical demonstrations would significantly strengthen the argument for the proposed methods' utility.
> > 2. The lack of experiments with larger models (e.g., 13B, 32B, or 72B parameters) limits the paper's claims on scalability. While computational constraints are understandable, even a partial analysis of scaling trends using available larger models or surrogate metrics would add critical value.
> > 3. There remains limited exploration of how the techniques could transfer to reasoning-heavy domains like physics or logic. Addressing this would enhance the generalizability and impact of the proposed methods.
> > 4. While the retrospective enhancement limitations were acknowledged, deeper analysis or proposed refinements to address these shortcomings (beyond focusing on tutorship amplification) would strengthen the paper's contribution to data synthesis methodologies.
> >
> > I appreciate the provided responses but encourage a more comprehensive engagement with these broader weaknesses to present a well-rounded evaluation of the paper's contributions and limitations.

---

> ### Author Response · Authors · 2024-12-02
> **Thank you for your response and clear suggestions for further updates**
>
> Thank you for your response and clear suggestions for further updates. The first point is a very direct and actionable suggestion, and we plan to directly follow it by adding illustrations for improvement; the second, third, and fourth points are extensions to different aspects of the paper, and the design and implementation of related experiments will take time. We currently offer some simple discussions and plans in response.
>
> First, we fully agree with the first point and consider adding illustrations in the first section to clarify the improvements brought by the methods. Due to the current revision limitations, here is a brief description of the process diagram: **Content**: Use problem-solving data directly to improve math reasoning skills instead of filtering the math corpus for mathematical knowledge expansion. **Highlight**: 1) Smaller data volume and higher training efficiency; 2) In further exploration, combining synthetic data methods, an iterative data-model update process is formed that combines model capability and data quality.
>
> For the second point, we believe that validating and providing larger models would be beneficial, but we will not repeat experiments on several research questions with larger models. Our current plans for data and models are to: 1) Implement quality identification and enhancement on synthetic data; 2) Conduct CPT with llama3 70b; 3) Implement post-training to apply inference stage techniques, allowing the problem-solving data construction and model training to form an iterative optimization. These attempts will take time to implement and improve based on actual improvements.
>
> For the third point, we have not validated the method's transfer due to the following two reasons: 1) We are not too concerned about the transfer of methods. Under the conclusions of the experiments in section 5.2, regarding the differences between physical reasoning and mathematical reasoning, I believe they are similar to the differences between high school and middle school math. It is relatively easy to implement experiments validating the model's transfer capabilities, and the results presented earlier have deepened our views. 2) In terms of expanding application value, expanding tasks including physics or logic is indeed what we hope for, but physical data is relatively scarce, and we have not yet begun to process the related raw data further, which will be a long-term process.
>
> For the fourth point, continuing the discussion from previous replies, we believe the intuitive assumption of retrospective enhancement is reasonable, but the data construction process does not reflect actual errors. A more direct correction of this method is to insert a potential error process in the current step instead of using subsequent steps to achieve the purpose of reflecting actual errors. To achieve this, the direct association is to use a method combining inference process and search, such as applying MCTS to build preferred data pairs, obtaining a correct inference path, obtaining another node with a large score gap based on an intermediate node, considered as a possible error process, and inserting it before the current node of the correct reasoning path, simulating "allowing models to retry upon regret". However, although this is a more direct refinement, it loses the original advantage of saving resources, so we did not optimize and implement this improvement idea at the design stage. Currently, we consider expanding based on small-scale trials to decide whether to expand.
>
> Thank you again for your response and suggestions for further updates and broader validations. The suggestions will be oriented towards practical implementation rather than further validation of the methods, proceeding according to the plan mentioned in the second point. For a deeper exploration of methods, we lean towards continuing exploration based on the ideas in 5.3, believing this to be important work, hoping to stimulate community interest in this exploration.

---

### Official Review · Reviewer_v6B9 · 2024-11-07

**Soundness:** 3
**Presentation:** 3
**Contribution:** 3
**Rating:** 6
**Confidence:** 4

**Summary:**

This paper aims to study the mathematical abilities of LLMs especially the effects of the type of training data (problem solving data vs general math data), using synthetic data in conjunction with real data, the effect of SFT vs CPT on the mathematical abilities of an LLM. The paper proposes training insights for optimizing the mathematical abilities of LLMs. The paper also proposes many synthetic data creation techniques like query expansion, retrospective enhancements, tutorship amplification for creating mathematical data. Finally they train a strong math specific model based on the findings of the previous questions.

**Strengths:**

- The paper is well written, and the overall direction of the paper is extremely important in the current times.
- Usually the data mixture ratio is one of the key secret sauces to training strong models, and the experiments in the initial section of the paper are important.
- The datasets used in the experiments are publicly available, and the experiments described in section 2 are extensive and cover the important cases.
- Synthetic data generation techniques can actually improve the robustness of the LLMs with respect to modifications (reframing of questions, or symbolic modifications) which could break the mathematical abilities of LLMs.

**Weaknesses:**

- For figure 1, have you tried ablating the percentage of the math mixture to extremes like 2:8, 1:9 or even complete problem solving data. For the tasks considered in the paper, it could be the case that simply doing CPT with the problem solving data could be sufficient.
- Overall I could not find much weakness in the paper, the experiments seem easy to understand and sound.

**Questions:**

- For response diversification, it could happen that the model could generate incorrect logic leading to the correct answer, do you have simple checks in place to ensure that this does not happen?
- For results in table 2, how will the results change if you apply inference time techniques like Best-of-N sampling or using a reward models during inference? It could be that the conclusion changes significantly. One recent work (https://arxiv.org/pdf/2410.02725) showed that simply restarting the inference can change the accuracy on math tasks significantly.

---

> ### Author Response · Authors · 2024-11-24
>
> Thank you for your review!
>
> ### Q1:  For figure 1, have you tried ablating the percentage of the math mixture to extremes like 2:8, 1:9 or even complete problem solving data. For the tasks considered in the paper, it could be the case that simply doing CPT with the problem solving data could be sufficient.
> **A1**: In the experimental design for Figure 1, we controlled the total amount of math data used in both the base group and the test groups to remain consistent, ensuring the full utilization of the data. Therefore, the math data mixture ratio was set at 3:7, with further discussion provided in the appendix under "Discussion on Experimental Setting."
>
> Adjusting the ratio to extremes would alter the actual amount of newly introduced data. However, the hypothesis you mentioned—that simply conducting CPT with problem-solving data could be sufficient—seems reasonable. To investigate this, we added an experimental group, Test4, which used only problem-solving data. The detailed results of this experiment are included in the newly added appendix, "Detailed Results of Problem-Solving Data Effectiveness Experiment."
>
> Overall, two conclusions emerged:
>
> 1. Test4 outperformed the experimental group with a math data mixture ratio of 3:7, highlighting the effectiveness of problem-solving data, which can even surpass the impact of adding a large number of new tokens.
> 2. When the math corpus is not used, the specific metrics of Test4 across the four evaluation datasets no longer align with those of Test1-3. This shift in data distribution undermines the improvement in GSM8K performance while enhancing the improvements in Math and Gaokao capabilities. From the perspective of ability dimensions, this may be due to the absence of the math corpus affecting the learning of general capabilities.
>
> Although this experimental group strengthens the conclusions, it seems to introduce additional variables, such as changes in data distribution. To maintain clarity in addressing this research question, we have included this experimental group in the appendix.
>
>
> ### Q2: For response diversification, it could happen that the model could generate incorrect logic leading to the correct answer. Do you have simple checks in place to ensure that this does not happen?
>
> **A2:** The process for response diversification, as well as the other three synthesis methods, follows the procedures described in the paper and is primarily based on prompt engineering. We did not introduce reward models or other process supervision mechanisms. Since response diversification  provided with process information of the seed data, occurrences of incorrect logic, while unavoidable, are significantly less severe compared to direct sampling followed by answer-based filtering.
>
> The seed data, i.e., the original problem-solving data, incorporates process supervision techniques, including sampling with various search methods and quality checks using PRM. However, we did not employ these methods during the data synthesis process due to their high computational cost. We plan to utilize such techniques after finalizing and optimizing the synthesis methods.
>
>
> ### Q3: For results in table 2, how will the results change if you apply inference time techniques like Best-of-N sampling or using a reward models during inference? It could be that the conclusion changes significantly. One recent work (https://arxiv.org/pdf/2410.02725) showed that simply restarting the inference can change the accuracy on math tasks significantly.
>
> **A3**: The mentioned inference-time techniques are indeed significant in the current field of mathematical reasoning. Best-of-N sampling is commonly combined with methods such as self-consistency, self-evaluative beam search, reasoning-via-planning, as well as techniques like Capability-Aware Self-Evaluations and Mid-Generation Self-Evaluations mentioned in the referenced paper, which directly improve Best-of-N sampling. However, regardless of the specific method employed, a common phenomenon typically holds: while the model's performance improves as the number of samples increases, if Model A is superior to Model B when sampling only one instance, Model A generally remains superior to Model B when sampling ten instances.
>
> Thus, we have two main concerns regarding this question. First, applying inference-time techniques in comparative experiments is rare. We could simply apply Best-of-N sampling and use metrics like pass@3 instead of pass@1 for evaluation, but it is difficult to anticipate that this would lead to a significant change in conclusions. Second, whether there is a specific connection between these techniques and the research question addressed in Table 2—namely, the impact of data distribution on the learning capabilities of CPT and SFT—or if this is merely an exploration of one experimental setting deemed important.
>
> Could you further clarify the main focus of this question? We could then conduct an additional experiments to address it.

---

> > ### Comment · Reviewer_v6B9 · 2024-11-28
> > **Response to the authors.**
> >
> > I am satisfied with the response and would like to stick to my score.

---

### Official Review · Reviewer_3zrA · 2024-11-07

**Soundness:** 3
**Presentation:** 4
**Contribution:** 3
**Rating:** 8
**Confidence:** 4

**Summary:**

This paper aims to improve mathematical reasoning capabilities in large language models, investigating whether problem-solving data can outperform general mathematical corpora in improving these capabilities during pre-training. The study explores three research questions: (1) effectiveness of problem-solving data, (2) data synthesis methods, and (3) comparison between pre-training (CPT) and supervised fine-tuning (SFT) stages. Results indicate that problem-solving data significantly improves reasoning performance, with the "tutorship amplification" data synthesis method proving particularly effective. The paper also introduces JiuZhang-8B, a model based on these findings which performs comparatively with other math LLMs.

**Strengths:**

* The paper nicely states the research questions. Also arranging results along the corresponding research question smoothly helps with reading the paper.
* The level of details in the dataset, data mixture, and experiments is very good.
* The introduced model JiuZhang-8B achieves competitive performance with fewer tokens, highlighting efficient training and well curated training tokens.
* The results are clear and I like the attempts to explain why behind the results, particularly the advantage of CPT over SFT for reasoning tasks.

**Weaknesses:**

* The paper shows promising results for mathematical reasoning but could benefit from testing on datasets that combine math with other fields, like physics. This would help assess if the model’s reasoning improvements generalize to interdisciplinary tasks requiring both math and domain knowledge, strengthening the case for broader applicability of the techniques.
* More analysis and clarity on why SFT underperforms CPT would strengthen the paper. Is the difference due to data volume limitations, catastrophic forgetting, or other factors?
* It would also be good to show some samples of the data, especially the generated or synthetic data. Also more details around data generation would be good like prompts or models used to generate data and if a human was used in the process for verification for example.

**Questions:**

* Is it really clear why SFT underperforms continual pretraining based on the experiments you performed?
* Does JiuZhang-8B learned math skills transfer to interdisciplinary tasks combining math with other fields?
* Could tutorship amplification be effective in other problem-solving domains?
* Is there any error analysis in JiuZhang-8B's problem-solving tasks?

---

> ### Author Response · Authors · 2024-11-23
> **Initial Response to Reviewer Comments (1)**
>
> Thank you for your review! Below are our responses to each comment:
>
> ### Q1: The paper shows promising results for mathematical reasoning but could benefit from testing on datasets that combine math with other fields, like physics. This would help assess if the model’s reasoning improvements generalize to interdisciplinary tasks requiring both math and domain knowledge, strengthening the case for broader applicability of the techniques.
>
> **A1**: Thank you for your suggestion!  We used the evaluation results of the mmlu-stem dataset from lm-evaluation-harness to demonstrate this point, and the results are presented in the table below. It is worth noting that the evaluation process determines outcomes based on the probabilities of the options, which might not fully represent a standard reasoning task. However, the results align with expectations, showing improvements in tasks somewhat related to mathematics. Specifically, among the subfields beyond mathematics, those more closely related—such as High School Statistics, College Chemistry, and Conceptual Physics—showed significant improvements.
>
> This is also consistent with our expectations. The results in Section 5.2 indicate that SFT and CPT primarily enhance capabilities aligned with their respective data distributions, and the performance in out-of-distribution (OOD) scenarios is actually influenced by the similarity of those tasks to the training data. Naturally, it can be inferred that for these non-mathematical tasks, the improvements may also be attributed to their relevance to mathematical tasks.
>
> Overall, generalization is indeed important and demonstrates a certain degree of broader applicability. However, I believe that identifying datasets that define related capabilities and using the approach proposed in the paper to transform them into in-distribution (IND) tasks could be a better option.
>
>
> |                | Stem | Elementary Mathematics | High School Statistics | College Chemistry | Conceptual Physics | Abstract Algebra |
> |----------------|------|------------------------|------------------------|-------------------|--------------------|------------------|
> | Meta-Llama-3-8B| 0.533143 | 0.4233             | 0.4815                 | 0.41              | 0.5319             | 0.33             |
> | JiuZhang-8B    | 0.588329 | 0.5741             | 0.5972                 | 0.51              | 0.6255             | 0.41             |
> | Difference     | 0.055200 | 0.1508             | 0.1157                 | 0.10              | 0.0936             | 0.08             |
>
> |                | High School Mathematics | High School Chemistry | High School Computer Science | College Computer Science | College Mathematics | Machine Learning |
> |----------------|-------------------------|-----------------------|------------------------------|--------------------------|---------------------|------------------|
> | Meta-Llama-3-8B| 0.3852                  | 0.5271                | 0.66                         | 0.50                     | 0.38                | 0.4286           |
> | JiuZhang-8B    | 0.4630                  | 0.6010                | 0.73                         | 0.54                     | 0.41                | 0.4554           |
> | Difference     | 0.0778                  | 0.0739                | 0.07                         | 0.04                     | 0.03                | 0.0268           |
>
> |                | Computer Security | High School Physics | College Physics | College Biology | High School Biology | Anatomy |
> |----------------|-------------------|---------------------|-----------------|-----------------|---------------------|---------|
> | Meta-Llama-3-8B| 0.77              | 0.4305              | 0.3824          | 0.7847          | 0.7484              | 0.6741  |
> | JiuZhang-8B    | 0.79              | 0.4437              | 0.3824          | 0.7569          | 0.7677              | 0.5852  |
> | Difference     | 0.02              | 0.0132              | 0.0000          | -0.0278         | 0.0193              | -0.0889 |

---

> > ### Author Response · Authors · 2024-11-23
> > **Initial Response to Reviewer Comments (2)**
> >
> > ### Q2: More analysis and clarity on why SFT underperforms CPT would strengthen the paper. Is the difference due to data volume limitations, catastrophic forgetting, or other factors?
> > **A2**: The related exploration is concentrated in Sections 5.2 and 5.3, where we divided the problem-solving data into subsets based on data distribution and difficulty level to investigate the sources of differences in learning mathematical capabilities between the CPT and SFT stages.
> > We did not delve deeper into the underlying reasons behind the related conclusions, and further discussions in follow-up work are undoubtedly important. Here are some of our intuitive thoughts:
> >
> >   - Regarding Result 4's mention of "both SFT and CPT primarily develop capabilities aligned with their data distributions", we speculate that the generalization ability of LLMs is limited. It might be confined to simple patterns such as numerical substitution and conditional result substitution. This limitation may depend on whether the training data includes similar data pairs for learning (akin to the issue of knowledge storage and extraction discussed in "Physics of Language Models: Part 3.1, Knowledge Storage and Extraction"). The generalization of reasoning capabilities warrants further experimental design and exploration.
> >
> >
> >    - Regarding Result 4's mention of "Regarding out-of-domain (OOD) capability learning, SFT is more susceptible to disturbances from data distribution compared to CPT," we speculate that the differences in OOD capability learning represented by different evaluation sets are influenced by the similarity of the evaluation set distribution to the training data; whereas the stability of CPT over SFT is due to the combined effects of generic data mixing and recall.
> >
> >   - For the main part of Result 4, it is noted that SFT's in-domain (IND) learning ability is weaker than that of CPT. This observation prompts the exploration and conclusions presented in Result 5.
> >
> >
> >    - Regarding Result 5, why hard training data results in learning capability differences between CPT and SFT is not well-understood, and may be suited for theoretical analysis, requiring meticulous work. Similar work from a theoretical analysis perspective seems to have just begun (Understanding Chair of That in LLMs Through Information Theory).
> >
> > ### Q3: It would also be good to show some samples of the data, especially the generated or synthetic data. Also, more details around data generation would be good, like prompts or models used to generate data and if a human was used in the process for verification, for example.
> > **A3**: Thank you for your suggestion. The four synthetic data generation methods we used do not involve human verification. To make this clear, we have added an appendix in the newly uploaded PDF, which provides the prompts used in the data synthesis process.
> >
> > ### Q4: Is it really clear why SFT underperforms continual pretraining based on the experiments you performed?
> > **A4**: This question aligns with Q2.
> >
> > ### Q5: Does JiuZhang-8B learned math skills transfer to interdisciplinary tasks combining math with other fields?
> > **A5**: This question aligns with Q1.
> >
> > ### Q6: Could tutorship amplification be effective in other problem-solving domains?
> > **A6**: We believe the effectiveness of tutorship amplification stems from both its viewpoint and process, which should be similar in other problem-solving domains. "Tutorship amplification" and "Retrospective Enhancement" share a similar perspective: models can identify errors but lack opportunities for correction. However, unlike "Retrospective Enhancement," which generates artificial errors that lead to sub-optimal results, "tutorship amplification" simulates a realistic error correction process. Thus, we believe that designing synthetic data methods tailored to the specific domain, based on this viewpoint, is crucial.
> >
> > ### Q7: Is there any error analysis in JiuZhang-8B's problem-solving tasks?
> > **A7**: We have not yet conducted a systematic error analysis of the evaluation set. We plan to design evaluation dimensions for error analysis of mathematical problems, particularly those with competition-level difficulty, after completing a series of post-training processes for JiuZhang-8B.

---

> > > ### Comment · Reviewer_3zrA · 2024-11-26
> > >
> > > Thank you for addressing the questions, clarifying some, and updating the paper. I will keep the score 8. This is a great paper and I wish you the best!

---

### Official Review · Reviewer_hewE · 2024-11-08

**Soundness:** 4
**Presentation:** 4
**Contribution:** 3
**Rating:** 8
**Confidence:** 3

**Summary:**

This work investigates strategies to improve mathematical reasoning in large language models (LLMs). It explores the effects of:

Problem-solving Data: Focuses on using math-specific problem-solving data during the pre-training phase, rather than general mathematical text, to enhance reasoning skills.

Data Synthesis Methods: Evaluates four synthesis methods—response diversification, query expansion, retrospective enhancement, and tutorship amplification—for creating synthetic problem-solving data to overcome data scarcity.

Training Stages: Compares the impact of applying problem-solving data during the continued pre-training (CPT) phase versus the supervised fine-tuning (SFT) stage.

Results suggest that problem-solving data enhances math reasoning more effectively than general data. The paper introduces JiuZhang-8B, a math-specific LLM trained with insights from these strategies, which reportedly achieves comparable or superior performance to existing math-focused models despite fewer training tokens.

There are several interesting takeaways. For example, even if one provides many hard problems in the training data, expect improvements primarily in the easy/medium regime. Overall, this work is a valuable contribution to the community.

**Strengths:**

This work tackles several research questions and addresses them clearly and effectively.

Specifically:

The study covers multiple aspects of training and data synthesis for LLMs in math reasoning, providing valuable insights into data handling (generation and sharding) strategies.

The analysis identifies tutorship amplification as particularly effective, offering practical value for generating synthetic data that improves reasoning abilities.

JiuZhang-8B demonstrates that the proposed methods lead to significant performance improvements, providing a practical model output as a proof of concept.

This work contributes to understanding how data complexity influences learning outcomes by segmenting data into difficulty levels and testing different mixtures.

**Weaknesses:**

While this paper summarizes several empirical results that are useful to the community, it makes limited efforts to advance understanding of the mechanics behind these observations. Future work in this direction would be of great value.

**Questions:**

Can the proposed synthetic data methods, especially "tutorship amplification", be effectively scaled or adapted to non-math domains?

How well would the identified CPT benefits translate to domains requiring other forms of reasoning (e.g., logical or scientific)?

Given the focus on hard data and high-volume problem-solving datasets, are there any recommendations for making this approach feasible for smaller labs with limited resources?

---

> ### Author Response · Authors · 2024-11-22
> **Initial Response to Reviewer Comments (1)**
>
> Thank you for your review! Below are our responses to each comment:
>
>
> ### Q1: While this paper summarizes several empirical results that are useful to the community, it makes limited efforts to advance understanding of the mechanics behind these observations. Future work in this direction would be of great value.
> **A1:** We agree with this point: advancing understanding of the mechanics behind these observations is an important direction. Specifically, any suggestions and speculations regarding the final result in section 5.3 on why hard training data leads to differences in learning ability between CPT and SFT are helpful. Intuitive conjectures behind other conclusions are discussed to some extent in the responses to the other reviewers, as follows:
>
>   - Regarding Result 4's mention of "both SFT and CPT primarily develop capabilities aligned with their data distributions", we speculate that the generalization ability of LLMs is limited. It might be confined to simple patterns such as numerical substitution and conditional result substitution. This limitation may depend on whether the training data includes similar data pairs for learning (akin to the issue of knowledge storage and extraction discussed in "Physics of Language Models: Part 3.1, Knowledge Storage and Extraction"). The generalization of reasoning capabilities warrants further experimental design and exploration.
>
>
>    - Regarding Result 4's mention of "Regarding out-of-domain (OOD) capability learning, SFT is more susceptible to disturbances from data distribution compared to CPT," we speculate that the differences in OOD capability learning represented by different evaluation sets are influenced by the similarity of the evaluation set distribution to the training data; whereas the stability of CPT over SFT is due to the combined effects of generic data mixing and recall.
>
>   - For the main part of Result 4, it is noted that SFT's in-domain (IND) learning ability is weaker than that of CPT. This observation prompts the exploration and conclusions presented in Result 5.
>
>
>    - Regarding Result 5, why hard training data results in learning capability differences between CPT and SFT is not well-understood, and may be suited for theoretical analysis, requiring meticulous work. Similar work from a theoretical analysis perspective seems to have just begun (Understanding Chair of That in LLMs Through Information Theory).
>
> ### Q2:  Can the proposed synthetic data methods, especially "tutorship amplification", be effectively scaled or adapted to non-math domains?
>
> **A2:** "Tutorship amplification" and "Retrospective Enhancement" are based on a similar viewpoint that models can detect errors but lack opportunities for correction. However, unlike "Retrospective Enhancement," which generates artificial errors leading to sub-optimal results, "tutorship amplification" simulates a realistic error correction process.  Therefore, we believe that designing synthetic data methods that are adapted to the corresponding domain based on this viewpoint is crucial.
>
> ### Q3: Given the focus on hard data and high-volume problem-solving datasets, are there any recommendations for making this approach feasible for smaller labs with limited resources?
> **A3:**  The main recommendation of the article is to shift sufficient problem-solving data to the CPT phase, replacing the need for multiple times the volume of generic mathematical corpora, which may not inherently be a research direction for smaller labs with limited resources. However, we would like to point out that for hard data, enhancing the quality and data synthesis efficiency through post-training and optimizing in conjunction with search algorithms during the inference process is crucial. This can be integrated with the capabilities developed during the CPT phase to form a multi-round iterative process, which could be an important direction for further enhancing mathematical reasoning abilities.

---

> > ### Comment · Reviewer_hewE · 2024-11-25
> > **Reply to the author's responses to my questions.**
> >
> > Thanks to the authors for the clarifications. These are quite satisfactory.

---

> ### Author Response · Authors · 2024-11-22
> **Initial Response to Reviewer Comments (2)**
>
> ### Q4: How well would the identified CPT benefits translate to domains requiring other forms of reasoning (e.g., logical or scientific)?
> **A4**: We illustrate our point using the results of the mmlu-stem subtasks, as shown in the table below. We observed improvements in high school statistics, college chemistry, and conceptual physics in subtasks outside of mathematics.
>
> This is also consistent with our expectations. The results in Section 5.2 indicate that SFT and CPT primarily enhance capabilities aligned with their respective data distributions, and the performance in out-of-distribution (OOD) scenarios is actually influenced by the similarity of those tasks to the training data. Naturally, it can be inferred that for these non-mathematical tasks, the improvements may also be attributed to their relevance to mathematical tasks.
>
> Overall, generalization is indeed important and demonstrates a certain degree of broader applicability. However, I believe that identifying datasets that define related capabilities and using the approach proposed in the paper to transform them into in-distribution (IND) tasks could be a better option.
>
>
> |                | Stem | Elementary Mathematics | High School Statistics | College Chemistry | Conceptual Physics | Abstract Algebra |
> |----------------|------|------------------------|------------------------|-------------------|--------------------|------------------|
> | Meta-Llama-3-8B| 0.533143 | 0.4233             | 0.4815                 | 0.41              | 0.5319             | 0.33             |
> | JiuZhang-8B    | 0.588329 | 0.5741             | 0.5972                 | 0.51              | 0.6255             | 0.41             |
> | Difference     | 0.055200 | 0.1508             | 0.1157                 | 0.10              | 0.0936             | 0.08             |
>
> |                | High School Mathematics | High School Chemistry | High School Computer Science | College Computer Science | College Mathematics | Machine Learning |
> |----------------|-------------------------|-----------------------|------------------------------|--------------------------|---------------------|------------------|
> | Meta-Llama-3-8B| 0.3852                  | 0.5271                | 0.66                         | 0.50                     | 0.38                | 0.4286           |
> | JiuZhang-8B    | 0.4630                  | 0.6010                | 0.73                         | 0.54                     | 0.41                | 0.4554           |
> | Difference     | 0.0778                  | 0.0739                | 0.07                         | 0.04                     | 0.03                | 0.0268           |
>
> |                | Computer Security | High School Physics | College Physics | College Biology | High School Biology | Anatomy |
> |----------------|-------------------|---------------------|-----------------|-----------------|---------------------|---------|
> | Meta-Llama-3-8B| 0.77              | 0.4305              | 0.3824          | 0.7847          | 0.7484              | 0.6741  |
> | JiuZhang-8B    | 0.79              | 0.4437              | 0.3824          | 0.7569          | 0.7677              | 0.5852  |
> | Difference     | 0.02              | 0.0132              | 0.0000          | -0.0278         | 0.0193              | -0.0889 |

---

### Official Review · Reviewer_awJ3 · 2024-11-08

**Soundness:** 3
**Presentation:** 3
**Contribution:** 3
**Rating:** 3
**Confidence:** 3

**Summary:**

The paper aims to understand how different aspects of training affect the ability of LLMs to do mathematical reasoning. This includes the tension between continuous pretraining and SFT, the use of problem solving data directly during CPT, the use of synthetic data generation and so forth. The authors provide a variety of fairly standard methods to test this, including some creative uses of synthetic data generation ( Query Expansion, Response Diversification, and Tutorship Amplification). They use LLAMA2 with the standard math evals, supplemented by a few that were introduced after the checkpoint in LLAMA2.  They apply their optimal methodologies to LLAMA3 to produce a new model JiuZhang-8B. JiuZhang-8B which has good performance especially relative to its size (Table 4)

**Strengths:**

The authors apply standard methods though in different variants than other work in this field. Their methods for synthetic data augmentation echo things I've seen before but not in this specific context.  They show some gains relative to larger models which is good.

**Weaknesses:**

This is solid work but does not feel like a major advance -- it is a set of good work with attention to evals and training procedures, but does not make any new conceptual advances.  IF ICLR were filled with papers like this one, the meeting wouldn't be intellectually exciting. This is very much "in the weeds" in engineering mixtures of LLM research

**Questions:**

What is the major advance of this paper?
What surprised you?
What did you learn that might generalize out of this narrow problem domain?

---

> ### Author Response · Authors · 2024-11-21
>
> ### Q：What is the major advance of this paper? What surprised you? What did you learn that might generalize out of this narrow problem domain?
> **A:** The results outlined in the Introduction Section hold both research and engineering value for the field of mathematical reasoning.
>
> **The major advances** include significant conclusions addressing three research questions and the development of the JiuZhang-8B model:
>
> - **Result 1 (RQ1):** Validates the efficiency of problem-solving data and motivates updates to math-specific model training paradigms. Specifically, it advocates for moving the more efficient problem-solving data to the CPT phase, replacing large volumes of math corpus.
> - **Result 2 (RQ2):** The depletion of internet data has made the exploration of synthetic data methods critically important. We validated the Tutorship Amplification method, which stands apart from commonly inspired synthetic data approaches such as Response Diversification and Query Expansion. This method leverages a teacher model to identify and correct errors based on the student model's responses, aiming to equip the model with self-correction capabilities. It has demonstrated significant performance improvements.
> - **Results 3–5 (RQ3):** Explore the distinction between CPT and SFT stages and identify the factors contributing to these differences. These findings enhance the understanding of how mathematical reasoning abilities are developed and emphasize the importance of preparing more challenging problem-solving data for the CPT phase.
>
> - **JiuZhang-8B** serves three primary purposes: to validate the research conclusions presented in the paper, to act as a base model that introduces a paradigm shift in training math-specific models with features worth further exploration, and to demonstrate competitive performance as a math-specific base model within 8B parameters.
>
> **This perspective could be tested in other domain-specific models.** Instead of merely providing domain-specific knowledge through corpora, introducing instruction tuning type data directly during CPT—which reflects the corresponding capabilities—might be the key to improving effectiveness. The process of defining necessary data and subsequently collecting or synthesizing it may surpass the approach of massively gathering internet corpora followed by quality and relevance filtering.
>
> **What surprised us** was the results regarding the difference in learning capabilities between CPT and SFT. This difference does not stem from CPT conferring out-of-distribution (OOD) capabilities, but rather from the impact of hard data within the training datasets. The underlying mechanisms behind this phenomenon merit further investigation.
>
> ---
>
> **Finally**, we would like to emphasize that conducting empirical research on the research questions based on current standard methods is not "in the weeds." On the contrary, we believe this approach provides empirical results that are significantly more meaningful compared to "conceptual" experiments, which start from a blank model and use only data related to the research questions. This is why we adopt various standard methods and begin exploring the research questions in a more complex and resource-intensive manner.
>
> We hope the results of this paper will influence updates to training paradigms and guide data collection and synthesis directions. The empirical findings and intermediate conclusions regarding differences introduced during the training phase aim to enhance the understanding of mathematical reasoning learning and inspire further exploration. The release of JiuZhang-8B serves as a revalidation of the research conclusions and, as a competitive math-specific base model, should be beneficial to the community.
>
> Additionally, we note that **Llemma** [1], which was accepted at ICLR 2024, primarily introduces the training results of a model along with some analyses. However, its empirical insights and the open-sourced model have effectively promoted subsequent explorations into LLM mathematical reasoning and have been widely cited.
>
> [1] Llemma: An Open Language Model For Mathematics

---

### Official Review · Reviewer_DwwE · 2024-11-08

**Soundness:** 1
**Presentation:** 1
**Contribution:** 2
**Rating:** 1
**Confidence:** 5

**Summary:**

This paper attempts to advance the mathematical reasoning capabilities of LLMs through  problem-solving data and exploring different data synthesis methods during pre-training and post-training stages. The authors propose findings on the efficacy of using problem-solving data and propose a new JiuZhang-8B model that outperforms baseline models.

**Strengths:**

Here are some strengths:
- I appreciate the clear introduction and quickly getting to the point on issues with the current two-stage paradigm.
- The authors include information for reproducibility, such as hyperparameters and data mixture ratios.
- The authors develop and release a new model, JiuZhang-8B, which could add value to the community.

**Weaknesses:**

This paper needs some work. Here is a list of areas for improvement:
- I recommend briefly describing in the abstract and introduction what you mean by problem-solving data.
- The paper outlines three research questions but fails to present a clear rationale or context for each. Why are these research questions important to answer?
- I find the bolding to be a bit hard to read in the introduction. I recommend laying out your contributions in a bulleted list to avoid having to bold “Result X”.
- I recommend providing a brief description of the techniques mentioned in Result 3 or instead summarizing what types of techniques are the most effective rather than specifying specific names.
- In the abstract, you say “improvements in mathematical reasoning achieved through continued pre-training (CPT) are often less significant compared to those obtained via SFT.” If this is true, then why do you say in result 4 that “While SFT can facilitate some learning of mathematical capabilities, it has a clear disadvantage compared to CPT”?
- I think listing Results 1 to 8 is ineffective as it is easy for the reader to get lost and hard to understand the overall bigger picture. I recommend consolidation the results by topic to ensure clarity.
- I recommend explaining why you chose the data mixture ratios that you did and moving the less important training details like batch size to the appendix.
- Adding figures throughout would greatly enhance the argument. For example, add a main pipeline figure of the overarching framework you are implementing, and potentially new figures for sections 3, 4, and 5.
- The methodology appears insufficiently rigorous, particularly in the way data distribution and difficulty levels are analyzed.
- The experiments lack comprehensive justification, and the comparisons between different training setups (CPT vs. SFT) come across as arbitrary. What are the hypotheses leading to these decisions?
- Throughout the paper, there are bold claims about the effectiveness of problem-solving data and specific synthesis methods like Tutorship Amplification. I recommend relying less on superficial observations and instead adding more deep analysis of why certain methods are effective.
- I recommend adding some theoretical backing or intuition as to why these methods work.
- The paper repeatedly discusses the distinction between CPT and SFT stages, yet the insights presented do not meaningfully advance the reader's understanding. I recommend reducing redundancy and focusing on meaningful results.
- I am concerned about data quality. It seems there is heavy reliance on synthetic data and vague descriptions of the datasets. What is the data cleansing process?
- The real-world applicability or significance of JiuZhang-8B is not discussed.


Minor:
- “We employed the MinHash deduplicationLee et al. (2022) framework to enhance training data quality by removing documents with significant duplicate content.” should have a space between “deduplication” and “Lee”

**Questions:**

I feel that a lot of my questions are already contained in the weaknesses above. Here are some more:
- How do you ensure representative and high-quality data?
- Can you provide more intuition or theoretical support for these empirical results?
- Beyond releasing JiuZhang-8B, what are the practical implications of this research?
- The paper makes significant claims about the advantages of CPT over SFT in developing reasoning capabilities. How do the authors justify these findings theoretically? Could these results be influenced by the specific model architecture or training configuration used?
- Is there a detailed error analysis provided for cases where the model underperforms, especially on more complex mathematical problems?

---

> ### Author Response · Authors · 2024-11-20
> **Initial Response to Reviewer Comments (1)**
>
> Thank you for your review! Below are our responses to each comment:
>
> ### Q1: I recommend briefly describing in the abstract and introduction what you mean by problem-solving data.
>
> **A1:** Thank you for the suggestion. We have included a brief description in the introduction. Problem-solving data refers to datasets composed of mathematical problems and their reasoning processes.
>
> ### Q2: The paper outlines three research questions but fails to present a clear rationale or context for each. Why are these research questions important to answer?
>
> **A2:** In the original text, the *Introduction* section provided the background and the key perspective of the paper, introducing the first research question (RQ1). RQ2 and RQ3 are extensions of RQ1. Additionally, at the beginning of Sections 3, 4, and 5, we explained their respective backgrounds and connections to earlier sections. Here, we revisit and elaborate on the rationale:
>
> We proposed **RQ1** based on two observations and a perspective:
> **RQ1:** *During the CPT stage, can providing problem-solving data more effectively enhance the model's mathematical reasoning capabilities compared to using general mathematical corpora?*
>
> - **Observation 1:** In math-specific LLM training under the current paradigm, the CPT stage often involves a large amount of filtered mathematical corpus tokens. However, the improvements in mathematical reasoning achieved through CPT are often less significant compared to those obtained via SFT.
> - **Observation 2:** Some other domains have attempted to introduce instruction tuning data to teach models how to utilize memorized knowledge during the pre-training stage.
> - **Perspective:** Due to the intrinsic distinction between mathematical knowledge and general world knowledge, different strategies are required for their effective acquisition and application. Mathematical knowledge involves a relatively limited set of elements, concepts, axioms, and theorems that need to be memorized and understood. The real challenge often lies not in recalling the relevant knowledge but in using this knowledge for reasoning or planning. Therefore, we propose that alternative strategies utilizing mathematical problems and their reasoning steps—referred to as problem-solving data—during the pre-training phase, to teach the model how to apply its memorized knowledge rather than simply increasing the volume of relevant data, can potentially lead to significant improvements in mathematical reasoning capabilities.
>
> Following the validation of the effectiveness of problem-solving data in RQ1, **RQ2** addresses the challenges arising from the limited availability of such data compared to internet-scale data. Specifically, it explores the necessity of efficient data synthesis methods and examines whether additional synthesis of problem-solving data during pre-training could further enhance model performance. This forms the basis of **RQ2:**
> **RQ2:** *If problem-solving data can enhance mathematical reasoning capabilities, are synthetic data from the same source equally effective, and what synthesis methods are most efficient?*
>
> Finally, **RQ3** seeks to resolve another dimension of RQ1. Specifically, it questions whether the effectiveness validated in RQ1 is solely due to the data itself or whether there are differences between developing mathematical reasoning skills during the CPT and SFT phases. Furthermore, if differences exist, **what factors contribute to them?** We believe that delving deeper into this distinction will help clarify the impact of training paradigms and provide valuable guidance for future data collection and synthesis efforts. This leads to **RQ3:**
> **RQ3:** *How does the effectiveness of problem-solving data during pre-training differ from its role in the SFT phase, and what factors contribute to these differences?*

---

> > ### Author Response · Authors · 2024-11-20
> > **Initial Response to Reviewer Comments (2)**
> >
> > ### Q3: I find the bolding to be a bit hard to read in the introduction. I recommend laying out your contributions in a bulleted list to avoid having to bold “Result X.”
> >
> > **A3:** Thank you for the suggestion. We have incorporated this and combined it with the changes suggested in Q6. The revisions have been included in the PDF and are summarized as follows:
> >
> > For **RQ1**, in Section 3, we compared the impact of using problem-solving data and explored various math data mixture ratios, leading to the following result:
> > - **Result 1**: Providing math problem-solving data significantly enhances the model's mathematical capabilities compared to general mathematical corpora, and a higher proportion of problem-solving data is more effective.
> >
> > For **RQ2**, in Section 4, we delved into four data synthesis techniques: response diversification, query expansion, retrospective enhancement, and tutorship amplification, leading to the following result:
> > - **Result 2**: Response diversification, query expansion, and tutorship amplification were effective. Among these, tutorship amplification methods emerged as distinctly superior, leveraging a teacher model to identify and correct errors based on the student model's responses, aiming to equip the model with self-correction capabilities.
> >
> > For **RQ3**, in Section 5.1, we first identified, from a holistic perspective, the differences in learning mathematical capabilities between the CPT and SFT stages using problem-solving data. Subsequently, in Sections 5.2 and 5.3, we divided the problem-solving data into subsets based on data distribution and difficulty level to investigate the sources of these differences:
> > - **Result 3**: Overall, while SFT can facilitate some learning of mathematical capabilities, it has a clear disadvantage compared to CPT.
> > - **Result 4**: From the perspective of data distribution, both SFT and CPT primarily develop capabilities aligned with their data distributions. However, SFT's in-domain (IND) learning ability is weaker than that of CPT. Regarding out-of-domain (OOD) capability learning, the conclusions are less clear, with only the observation that SFT is more susceptible to disturbances from data distribution compared to CPT.
> > - **Result 5**: From the perspective of difficulty level, providing hard multi-step problem-solving data enables more effective learning, with this advantage being particularly evident in CPT compared to SFT. This may be the primary source of the learning capability differences between CPT and SFT. Therefore, we recommend preparing more challenging problem-solving data for the CPT phase.
> >
> > ### Q4: I recommend providing a brief description of the techniques mentioned in Result 3 or instead summarizing what types of techniques are the most effective rather than specifying specific names.
> >
> > **A4:** Thank you for the suggestion. We have added a brief description:
> > Tutorship amplification methods emerged as distinctly superior, leveraging a teacher model to identify and correct errors based on the student model's responses, aiming to equip the model with self-correction capabilities.
> >
> > ### Q5: In the abstract, you say, “improvements in mathematical reasoning achieved through continued pre-training (CPT) are often less significant compared to those obtained via SFT.” If this is true, then why do you say in Result 4 that “While SFT can facilitate some learning of mathematical capabilities, it has a clear disadvantage compared to CPT”?
> >
> > **A5:** This question arises from a misunderstanding. The full sentence in the abstract is:
> > *"Despite these efforts, the improvements in mathematical reasoning achieved through continued pre-training (CPT) are often less significant compared to those obtained via SFT."*
> > This statement, following the introduction of the state of current math-specific models, is a summary of the current state of research. It aims to provide background, introduce our motivation, introduce our viewpoints and research questions, as already reiterated in **A2**.
> >
> > ### Q6: I think listing Results 1 to 8 is ineffective as it is easy for the reader to get lost and hard to understand the overall bigger picture. I recommend consolidating the results by topic to ensure clarity.
> >
> > **A6:** Thank you for the suggestion. We have addressed this together with Q3. Please refer to **A3** for the revised structure.

---

> > > ### Author Response · Authors · 2024-11-20
> > > **Initial Response to Reviewer Comments (3)**
> > >
> > > ### Q7: I recommend explaining why you chose the data mixture ratios that you did and moving the less important training details like batch size to the appendix.
> > >
> > > **A7:** Thank you for the suggestion. Together with Q10, we have added an appendix titled *Discussion on Experimental Setting* to explain the rationale behind the experimental setup details in each section. Below is the content of the appendix:
> > >
> > > ---
> > >
> > > ### Appendix: Discussion on Experimental Setting
> > >
> > > Our experimental design generally adheres to the principles of comparative experiments, forming control groups to test hypotheses by introducing variations. Below, we elaborate on the considerations behind the design of each experimental group.
> > >
> > > - **Section 3:** To mitigate the influence of the total amount of math data used on the experimental conclusions, as described in the main text, we controlled the total amount of math data used in both the base group and the test groups to be the same. Specifically, the base group used the entire 14.7B math corpus as the math data, while for the test groups, we used 7.2B problem-solving data and split the remaining 7.5B math corpus to ensure that the total math data used was also 14.7B.
> > >   Subsequently, in three test groups, we adjusted the math data mixture ratio to further verify the effectiveness of problem-solving data and determine the optimal mixture ratio for subsequent experiments. Specifically:
> > >   - Starting with a math data mixture ratio of **5:5**.
> > >   - Adjusting to **3:7**, where the smaller ratio data, 7.5B math corpus, could be fully utilized within 10K steps and used more than twice within 25K steps, ensuring full data utilization.
> > >   - Finally, conducting a complementary experiment with a reverse ratio of **7:3**.
> > >
> > > - **Section 4:** To delineate the impact of different synthetic data, we introduced a control group, Base2, which used the entire problem-solving data on top of Base1. The experimental group further incorporated synthetic data into this setup to verify that the synthetic data contributed new value, rather than just the ability of the original data.
> > >
> > > - **Section 5.1:** We compared how the stage at which problem-solving data is introduced (CPT vs. SFT) significantly affects the model’s ultimate capabilities.
> > >   - Following the setups of Base1 and Base2, we conducted SFT using the same data on Base1 to create a comparative experiment. We hypothesized that Base1-SFT would benefit from enhanced instruction-following ability, which Base2 might lack.
> > >   - To validate this, we partitioned **1% of the data**, assuming it had limited impact on reasoning ability but contributed to instruction-following ability. This 1% data was applied for SFT on both Base1 and Base2 groups.
> > >   - By comparing **Base1-1%SFT with Base1-SFT**, we evaluated the reasoning ability gained from SFT. Similarly, by comparing **Base1-1%SFT with Base2-1%SFT**, we assessed the reasoning ability gained from CPT.
> > >
> > > - **Section 5.2:** Base1 was reused to define the improvement in the experimental groups’ abilities. We focused on differences in capabilities across the same evaluation dataset representing the data distribution at various training stages.
> > >   - Two experimental groups, **Middle-school-SFT** and **Middle-school-CPT**, were introduced, using a Middle-school data subset from the training set for SFT and CPT, respectively. This allowed us to evaluate the in-domain (IND) learning differences between SFT and CPT on a specific evaluation dataset.
> > >   - Differences in out-of-domain (OOD) learning on other evaluation datasets were also analyzed.
> > >   - To strengthen the robustness of the conclusions, the Middle-school subset was replaced with a **High-school subset**, and the same experiments were repeated.
> > >
> > > - **Section 5.3:** The experimental design was similar to that in Section 5.2, with a key difference: we focused on variations in learning ability of training data with different difficulty levels at different training stages.
> > >   - Besides comparing SFT and CPT using **easy** and **hard** subsets of the training data separately, we also contrasted the performance of different training subsets within the same training stage.
> > >   - Comparisons based on evaluation dataset distributions were used only as supplementary analysis.
> > >
> > > ---

---

> ### Author Response · Authors · 2024-11-20
> **Initial Response to Reviewer Comments (4)**
>
> ### Q8: Adding figures throughout would greatly enhance the argument. For example, add a main pipeline figure of the overarching framework you are implementing, and potentially new figures for Sections 3, 4, and 5.
>
> **A8:** Thank you for the suggestion. We initially considered adding explanatory diagrams for the four methods described in Section 4 to improve clarity, but this was limited by space constraints.
>
> ---
>
> ### Q9: The methodology appears insufficiently rigorous, particularly in the way data distribution and difficulty levels are analyzed.
>
> **A9:** Thank you for the suggestion. This issue have been discussed in detail in the response to reviewer fBsg. Some clarifications and additional analyses have already been included in Appendix B of the revised PDF.
>
> ---
>
> ### Q10: The experiments lack comprehensive justification, and the comparisons between different training setups (CPT vs. SFT) come across as arbitrary. What are the hypotheses leading to these decisions?
>
> **A10:** We have reiterated the background of our research questions in **A2** and discussed the experimental setup rationale in detail in **A7**. These provide clear hypotheses and motivations for the experimental comparisons.
>
> ---
>
> ### Q11: Throughout the paper, there are bold claims about the effectiveness of problem-solving data and specific synthesis methods like Tutorship Amplification. I recommend relying less on superficial observations and instead adding more deep analysis of why certain methods are effective.
>
> **A11:** Thank you for the suggestion. We have addressed this issue in **A12**, where we provide a detailed discussion of all the results.
>
> ---
>
> ### Q12: I recommend adding some theoretical backing or intuition as to why these methods work.
>
> **A12:** Thank you for the suggestion.We did not conduct a theoretical analysis. The following is our discussion on intuition from five conclusions.
>
> 1. **Result 1 (Problem-solving data effectiveness):**
>  As outlined in the *Introduction*, the real challenge often lies not in recalling the relevant knowledge but in using this knowledge for reasoning or planning. Therefore, we should use problem solving data instead of math corpus.
>
> 2. **Result 2 (Tutorship Amplification):**
>    The effectiveness of Tutorship Amplification can be attributed to its ability to provide synthetic data that imparts self-correction capabilities to the model. By leveraging a teacher model to identify and correct errors in the student model’s responses,  the data generated by this method endows the model with the ability to detect and correct errors, thereby enhancing its reasoning ability.
> 3. **Results 3–5 (Differences between CPT and SFT):**
>    These findings stem from RQ3: *How do the capabilities developed from the same problem-solving data differ between the CPT and SFT stages, and what factors contribute to these differences?*
> The experimental results should be reliable and provide valuable insights for further analysis. We did not delve deeper into this matter, and further discussions on the follow-up work are undoubtedly important. Here are some of our intuitive thoughts:
>
>   - Regarding Result 4's mention of "both SFT and CPT primarily develop capabilities aligned with their data distributions", we speculate that the generalization ability of LLMs is limited. It might be confined to simple patterns such as numerical substitution and conditional result substitution. This limitation may depend on whether the training data includes similar data pairs for learning (akin to the issue of knowledge storage and extraction discussed in "Physics of Language Models: Part 3.1, Knowledge Storage and Extraction"). The generalization of reasoning capabilities warrants further experimental design and exploration.
>
>
>    - Regarding Result 4's mention of "Regarding out-of-domain (OOD) capability learning, SFT is more susceptible to disturbances from data distribution compared to CPT," we speculate that the differences in OOD capability learning represented by different evaluation sets are influenced by the similarity of the evaluation set distribution to the training data; whereas the stability of CPT over SFT is due to the combined effects of generic data mixing and recall.
>
>   - For the main part of Result 4, it is noted that SFT's in-domain (IND) learning ability is weaker than that of CPT. This observation prompts the exploration and conclusions presented in Result 5.
>
>
>    - Regarding Result 5, why hard training data results in learning capability differences between CPT and SFT is not well-understood, and may be suited for theoretical analysis, requiring meticulous work. Similar work from a theoretical analysis perspective seems to have just begun (Understanding Chair of That in LLMs Through Information Theory).

---

> ### Author Response · Authors · 2024-11-20
> **Initial Response to Reviewer Comments (5)**
>
> ### Q13: The paper repeatedly discusses the distinction between CPT and SFT stages, yet the insights presented do not meaningfully advance the reader's understanding. I recommend reducing redundancy and focusing on meaningful results.
>
> **A13:** Thank you for the suggestion. We have streamlined the discussion of the distinction between CPT and SFT stages and consolidated the key insights into three main results to avoid redundancy while emphasizing meaningful findings:
>
> 1. **Result 3 (CPT vs. SFT learning capability differences):**
>    This result validates the differences in learning capabilities between CPT and SFT. Combined with **Result 1**, it provides actionable guidance for updating training paradigms in math-specific models. Specifically, it supports the strategy of moving sufficient problem-solving data to the CPT phase, replacing multiple times the volume of generic math corpus.
>
> 2. **Result 4 (CPT’s robustness to OOD capabilities):**
>    This result demonstrates that CPT's OOD capability is less affected by data distribution disturbances compared to SFT. It also confirms that introducing training data aligned with the target distribution remains the optimal choice for improving specific capabilities.
>
> 3. **Result 5 (Primary source of CPT and SFT differences):**
>    This result identifies the primary source of learning capability differences between CPT and SFT. It emphasizes the importance of preparing more challenging problem-solving data for the CPT phase.
>
> ### Q14: I am concerned about data quality. It seems there is heavy reliance on synthetic data and vague descriptions of the datasets. What is the data cleansing process?
>
> **A14:** The methods for data synthesis have been detailed in Section 4 of the paper. The deduplication and depollution processes for the seed dataset used in synthesis are discussed in Section 2. Ensuring the quality of the seed dataset is a comprehensive task, involving both general techniques such as quality filtering and relevance filtering, as well as the integration and iterative refinement of reward models with reasoning processes.
>
>
> ### Q15: The real-world applicability or significance of JiuZhang-8B is not discussed.
>
> **A15:** JiuZhang-8B serves three primary purposes:
> 1. To validate the research conclusions presented in the paper.
> 2. To act as a foundational model that introduces a paradigm shift in training math-specific models, with features worth further exploration.
> 3. To demonstrate competitive performance as a math-specific base model within 8B parameters.
>
> ### Q16: How do you ensure representative and high-quality data?
>
> **A16:** This concern overlaps with **Q14**.
>
>
> ### Q17: Can you provide more intuition or theoretical support for these empirical results?
>
> **A17:** This question aligns with **Q12**.

---

> > ### Author Response · Authors · 2024-11-23
> > **Initial Response to Reviewer Comments (6)**
> >
> > ### Q18: Beyond releasing JiuZhang-8B, what are the practical implications of this research?
> >
> > **A18:** We believe the three research questions we proposed, along with their corresponding conclusions, are of significant importance. One function of JiuZhang-8B is to validate these conclusions.
> > - **Result 1 (RQ1):** Validates the efficiency of problem-solving data and motivates updates to math-specific model training paradigms. Specifically, it advocates for moving the more efficient problem-solving data to the CPT phase, replacing large volumes of math corpus.
> > - **Result 2 (RQ2):** The depletion of internet data has made the exploration of synthetic data methods critically important. We validated the Tutorship Amplification method, which stands apart from commonly inspired synthetic data approaches such as Response Diversification and Query Expansion. This method leverages a teacher model to identify and correct errors based on the student model's responses, aiming to equip the model with self-correction capabilities. It has demonstrated significant performance improvements.
> > - **Results 3–5 (RQ3):** Explore the distinction between CPT and SFT stages and identify the factors contributing to these differences. These findings enhance the understanding of how mathematical reasoning abilities are developed and emphasize the importance of preparing more challenging problem-solving data for the CPT phase.
> >
> >
> >
> > ### Q19: The paper makes significant claims about the advantages of CPT over SFT in developing reasoning capabilities. How do the authors justify these findings theoretically? Could these results be influenced by the specific model architecture or training configuration used?
> >
> > **A19:** We did not conduct formal theoretical analysis to justify our research questions. However, our conclusions are based on rigorous comparative experiments. If stronger open-source base models become available in the future, we will consider validating our findings with them to further strengthen the evidence.
> >
> >
> > ### Q20: Is there a detailed error analysis provided for cases where the model underperforms, especially on more complex mathematical problems?
> >
> > **A20:** Currently, we have not evaluated the model on competition-level mathematical problems. We plan to focus on such problems and conduct a detailed error analysis after completing a series of post-training tasks on JiuZhang-8B. This will help identify and address areas for performance improvement.

---

> ### Author Response · Authors · 2024-12-02
> **Have we addressed your concern?**
>
> Thank you very much for your thorough review. As the discussion period is nearing its conclusion, I wanted to follow up to ensure you’ve had the opportunity to review our detailed rebuttal. Given the additional explanations and adjustments we've incorporated, we would greatly appreciate your feedback on whether our responses have satisfactorily addressed your concerns.
>
> Thank you once again for your time and thoughtful review. We look forward to your response.

---

> > ### Comment · Reviewer_DwwE · 2024-12-02
> > **Response 1**
> >
> > I appreciate the authors’ thorough response to my concerns. While your revisions and additional explanation help clarify the motivation behind each research question, I am still concerned about the coherence and narrative of the paper. The framing of your research questions gives the impression of addressing a checklist of inquiries rather than telling a cohesive story relevant to existing literature. A strong ICLR paper should present a clear narrative that ties to all research questions together under a unified theme or framework. I have rarely seen such a format of lists of questions and results. While you describe how RQ2 and RQ3 extend RQ1, the paper does not sufficiently articulate how these extensions flow logically and conceptually. For example, why is it important to understand synthetic data (RQ2) as an extension of problem-solving data in RQ1? Similarly, for RQ3, what broader implications does the CPT vs. SFT distinction have for advancing our understanding of mathematical reasoning or pre-training paradigms? These connections need to be more explicitly developed. The rationale for the research questions remains overly focused on the internal logic of the study rather than the broader significance for the field.
> >
> > Moreover, while your response mentions that sections 3, 4, and 5 explain the backgrounds of the respective research questions, the paper should aim for tighter integration between these sections. It should not merely provide localized context but build a cohesive argument that develops across the paper. For instance, how do the findings of RQ1 directly set up the need for RQ2, and how does the discussion of RQ2 naturally lead into the exploration of RQ3?
> >
> > Thank you for your detailed explanation in the appendix. I still find the response unsatisfactory for the following reasons. While you describe the specific mixture ratios used (e.g., 5:5, 3:7, and 7:3), the rationale provided for these choices remains vague. For instance, why were these particular ratios selected? What theoretical or empirical insights guided these choices, and how do they connect to your research objectives or hypotheses? The explanation seems to focus more on the mechanics of implementation rather than on why these specific settings were expected to yield meaningful insights. It almost feels that these mixtures were chosen just to help your paper.
> >
> > I disagree that you were limited by space for adding figures. Many papers have over 5 figures, and this can be easily accounted for by appropriately moving items around in LaTeX. Figures are vital for understanding; a block of text is not always necessary and limits readability.
> >
> > While I appreciate the additional details and the references to Appendix B, I remain unsatisfied with how the paper addresses the rigor of data distribution and difficulty level analysis. What specific statistical or experimental procedures were employed to assess the alignment between training data and evaluation data distributions? How were the difficulty levels of the data quantitatively defined and validated? Was this categorization based on established benchmarks, or was it purely heuristic? Moreover, observing the correlation between data difficulty and performance improvements does not offer deeper insights or novel contributions specific to mathematics.
> >
> > Crucially, while the results highlight differences in learning capabilities between CPT and SFT phases, there is limited discussion on the trade-offs involved. What are the computational costs and scalability implications of focusing on high-difficulty data subsets? Moreover, a rigorous analysis of data distribution and difficulty levels is central to evaluating the validity of the methodology and should not be relegated to supplementary material.
> >
> > Referring back to A2 and A7 does not resolve the issue of arbitrariness. The response appears to sidestep the need for a direct and robust explanation of how the experiments were designed to test well-defined hypotheses.

---

> > ### Comment · Reviewer_DwwE · 2024-12-02
> > **Response 2**
> >
> > While you claim that A12 provides detailed discussions of all results, the explanations remain surface-level and do not probe the underlying mechanisms or theoretical justifications. Result 1 states that problem-solving data is effective because it focuses on reasoning and planning rather than recall, but this merely reiterates your premise without analyzing why this distinction enhances performance. Result 2 attributes the success of Tutorship Amplification to self-correction capabilities, but this is an effect, not an analysis of the underlying reason why the method produces superior results. The reliance on intuition rather than robust analysis or theoretical backing makes the claims feel speculative. Why does Tutorship Amplification succeed where other synthesis methods fail? Is it due to the quality of corrections, diversity of generated data, or some other factor? This is not explored in depth. What features of problem-solving data distinguish it from mathematical corpora in facilitating reasoning? Could there be unintended biases in the data or training process that contribute to the observed results? Stating that no theoretical analysis was conducted and deferring it to future work undermines the credibility of the bold claims made in the paper. While not every paper needs a rigorous theoretical framework, there should at least be an attempt to provide plausible, evidence-backed hypotheses for why these methods work. The “intuition” provided for Results 3–5 is vague and speculative. The claim that “generalization ability of LLMs is limited to simple patterns” is not substantiated with clear evidence or citations. If this is a hypothesis, it should be validated through specific experiments or referenced in existing work. The statement that “SFT is more susceptible to disturbances due to the similarity of evaluation set distributions” lacks a concrete explanation of why this would occur or how it connects to the training dynamics of SFT versus CPT.
> >
> > In Section 2, you describe the use of MinHash deduplication to remove duplicate documents and contaminated data from OpenWebMath. There is no mention of the specific thresholds or criteria used for deduplication and decontamination. For example, what constitutes “significant duplicate content,” and how were contamination risks quantitatively assessed? Were there any manual quality checks or other validation methods applied post-decontamination? What metrics or benchmarks were used to evaluate the correctness, diversity, and relevance of the synthetic data? How was the "completeness of data" operationalized and measured? Were the reasoning steps evaluated manually or via automated processes? If automated, what mechanisms ensured the accuracy of this categorization?
> >
> > However, the explanation for the real-world applicability or significance of JiuZhang-8B remains inadequate and misses the point of the critique. The response merely reiterates the goals stated in the paper without addressing how JiuZhang-8B contributes meaningfully to real-world scenarios or the broader field of mathematics-specific language models. Claiming that JiuZhang-8B "validates the research conclusions" is circular reasoning. This does not address its significance beyond the scope of your own study. The mention of a “paradigm shift” is vague and unsubstantiated. How does JiuZhang-8B specifically shift paradigms in training math-specific models, and why is this significant?
> >
> > Stating that the conclusions could be validated with stronger open-source models in the future shifts responsibility away from this work. A meaningful paper should stand on its own by addressing potential confounding factors, such as architecture and training setup, within the current scope. The absence of a detailed error analysis severely limits the paper's practical relevance and the interpretability of its results. Complex problems are where the strengths and weaknesses of mathematical reasoning models are most apparent. Without testing on such problems, it is unclear how your findings translate to real-world or high-stakes scenarios.

---

> > ### Comment · Reviewer_DwwE · 2024-12-02
> > **Response 3**
> >
> > Overall, the response focuses almost entirely on internal validation of research questions and methodology. While this is valuable for the study's internal coherence, it does not address how the findings translate to practical applications or advancements in the field. Simply stating that the results “enhance understanding” is not enough. What actionable steps or innovations could arise from this research? While the theoretical insights into CPT and SFT are valuable, the practical benefits are not well-articulated. How could these insights inform the design of training strategies for next-generation LLMs? Are there implications for cost, scalability, or accessibility in training models?
> >
> > After carefully considering the revised submission and responses provided, I must lower my score for this paper from 3 to 1. My decision is primarily driven by the persistent lack of rigor and coherence in addressing key concerns, particularly given the time afforded for revision. The authors’ responses repeatedly fall short of expectations for a high-quality submission to ICLR. Overall, the revisions do not demonstrate sufficient progress or rigor to address the substantial concerns raised, and I am unsatisfied that this is the best that could be achieved in the time provided. As a result, I cannot recommend this paper for acceptance.

---

> ### Author Response · Authors · 2024-12-04
> **The Main Objection in Response 3 is Unreasonable**
>
> Firstly, we would like to thank the reviewer for their response, which includes clarifying concerns about previous issues and summarizing their opinions.
>
> We would like to focus on **the summary objection** raised in Response 3. The reviewer’s reason for rejecting the submission is **“unsatisfied that this is the best that could be achieved in the time provided.”** We would like to clarify that we already provided a detailed response **15 days ago**, which included answers to the 20 raised questions, significant revisions to the paper, and additional appendices. Given that most of the questions were succinct (often within a line), we made further expansions and discussions on certain points.  However, the reviewer only provided additional clarifications on some of the concerns on **the last day** of the discussion period. Clearly, we could not have anticipated and addressed the unexpressed concerns beforehand. Thus, using “unsatisfied that this is the best that could be achieved in the time provided” as the core reason for rejection is manifestly unfair.
>
> **The review comments on the paper itself should serve as the basis for judgment, yet the core objection raised by the reviewer remains uncertain.** For the reasons for their decision, the reviewer stated, "My decision is primarily driven by the persistent lack of rigor and coherence in addressing key concerns." However, at the beginning of Response 3, the reviewer, in reference to our initial response, remarked, "While this is valuable for the study's internal coherence, it does not address how the findings translate to practical applications or advancements in the field." This contradictory and unclear core objection reinforces our position that the main objection in Response 3 is unreasonable. As for the mentioned issue of "does not address how the findings translate to practical applications," the reviewer elaborated that this concern arises from the statement, "While the theoretical insights into CPT and SFT are valuable, the practical benefits are not well-articulated." This implies that in our design of comparative experiments to validate and discuss RQ3, we should address the practical benefits. This is illogical, as introducing additional considerations of practical benefits in strictly controlled experimental comparisons could introduce potential confounding factors, thereby undermining the validation of our conclusions.
>
> In fact, **we have already provided multiple explanations regarding the motivation of the paper, the theoretical insights offered, and the practical benefits in both the paper and the Initial Response (A2, A3, A12, A13, A15, A18).** A summary of these points is as follows:
>
> ***We proposed RQ1 **[1]** based on two observations and a perspective. We then extended RQ1 to RQ2 **[2]**  and RQ3 **[3]** by considering the limitations in problem-solving data quantity and the potential learning ability differences due to the training phase. Finally, based on the research findings, we developed the competitive math-specific base model JiuZhang-8B, which, as a comprehensive validation of the research conclusions, brings about a paradigm shift in training math-specific models **[4]** .***
>
> **[1] RQ1:** *During the CPT stage, can providing problem-solving data more effectively enhance the model's mathematical reasoning capabilities compared to using general mathematical corpora?*
>
> - **Observation 1:** In math-specific LLM training under the current paradigm, the CPT stage often involves a large amount of filtered mathematical corpus tokens. However, the improvements in mathematical reasoning achieved through CPT are often less significant compared to those obtained via SFT.
> - **Observation 2:** Some other domains have attempted to introduce instruction tuning data to teach models how to utilize memorized knowledge during the pre-training stage.
> - **Perspective:** Due to the intrinsic distinction between mathematical knowledge and general world knowledge, different strategies are required for their effective acquisition and application. Mathematical knowledge involves a relatively limited set of elements, concepts, axioms, and theorems that need to be memorized and understood. The real challenge often lies not in recalling the relevant knowledge but in using this knowledge for reasoning or planning. Therefore, we propose that alternative strategies utilizing mathematical problems and their reasoning steps—referred to as problem-solving data—during the pre-training phase, to teach the model how to apply its memorized knowledge rather than simply increasing the volume of relevant data, can potentially lead to significant improvements in mathematical reasoning capabilities.

---

> > ### Author Response · Authors · 2024-12-04
> > **Supplement to the previous section**
> >
> > **[2] RQ2**: *If problem-solving data can enhance mathematical reasoning capabilities, are synthetic data from the same source equally effective, and what synthesis methods are most efficient?*
> >
> > RQ2 is an extension of RQ1 from an applied perspective. The limited availability of problem-solving data, compared to internet-scale data, is a major barrier to replacing math corpora with problem-solving data. RQ2 seeks to verify the effectiveness of synthetic data and identify efficient synthesis methods, thereby strengthening the feasibility of applying the conclusions from RQ1 to practical scenarios.
> >
> > **[3] RQ3**: *How does the effectiveness of problem-solving data during pre-training differ from its role in the SFT phase, and what factors contribute to these differences?*
> >
> > RQ3 seeks to address an extended issue naturally arising from RQ1: whether the use of problem-solving data in RQ1 could still be relevant in the SFT phase. Then, RQ3 questions whether the effectiveness validated in RQ1 is due solely to the data itself or whether there are differences in developing mathematical reasoning skills between the CPT and SFT phases. This leads to a revision of the training paradigm, suggesting that moving problem-solving data from the SFT phase to the CPT phase is beneficial and enhances efficiency.
> >
> > After validating the impact of the training phases through simple comparative experiments, we aim to explore what factors contribute to these differences. Sections 5.2 and 5.3 delve into the dimensions of data distributions and difficulty levels, concluding that providing hard multi-step problem-solving data enables more effective learning, with this advantage being particularly evident in CPT compared to SFT. We believe this conclusion provides theoretical insights that encourage further exploration and also provides practical suggestions for data acquisition directions.
> >
> >
> > **[4]**  A paradigm shift in training math-specific models refers to acquiring more problem-solving data and moving it from the SFT phase to the CPT phase, directly targeting the learning of reasoning skills. This shift replaces the current practice of focusing the CPT phase on collecting internet-scale data and filtering large amounts of mathematical corpora for learning math knowledge. This change leads to a significant improvement in efficiency. Furthermore, combining with synthetic data methods, it can be generalized into an iterative data-model update process, where model capabilities and data quality alternately improve.

---

> ### Author Response · Authors · 2024-12-04
> **Response to questions after clarifying concerns in Response 1**
>
> Next, we will provide specific responses to the questions following the clarifying concerns. It is important to note that, despite adding further descriptions, some of the reviewer's questions still appear to be more like expressions of vague impressions rather than concrete issues, making the actual questions somewhat confusing.
>
> **The reviewer initially expressed dissatisfaction with the overall story of the paper.** However, we have already provided explanations from two aspects: the motivation and the importance of the three research questions. In the earlier responses, the reviewer seemed to be caught in a dilemma—while they claimed that the motivation was not clearly articulated, they also expressed concerns about the coherence between the three research questions.
>
> We retell the complete story again: We proposed RQ1 [1] based on two observations and a perspective. We then extended RQ1 to RQ2 [2] and RQ3 [3] by considering the limitations in problem-solving data quantity and the potential learning ability differences arising from the training phase. Finally, based on the research findings, we developed the competitive math-specific base model, JiuZhang-8B, which, as a comprehensive validation of our research conclusions, brings about a paradigm shift in training math-specific models [4].  References can be found in the previous sections.
>
> **The reviewer next expressed concerns regarding the design details of the mixture ratios.**
>
> The different math data mixture ratios used in the control groups provided validation for Result 1, specifically the finding that "Providing math problem-solving data significantly enhances the model’s mathematical capabilities compared to general mathematical corpora," as well as the discovery that "a higher proportion of problem-solving data is more effective."
>
> We have already addressed the question Q7, "Why did you choose these data mixture ratios?" in A7 and the Appendix: Discussion on Experimental Setting. Regarding the newly raised question, "Why were these specific settings expected to yield meaningful insights?", we argue that the Result drawn from the control experiments themselves constitute the meaningful insights.
>
> Furthermore, the reviewer’s specific query in v6B9, “Why not ablate the percentage of the math mixture to extremes like 2:8, 1:9, or even complete problem-solving data?” was addressed in our response. We explained the rationale behind not using extreme settings  in the response and supplemented this with additional experiments, which were included in the Appendix under Detailed Results of Problem-Solving Data Effectiveness Experiment.
>
> **The reviewer next expressed dissatisfaction with the absence of additional figures.**
>
> The original comment raised an odd request—adding figures simply for the sake of adding them. In response, we explained that we had initially considered including a figure to illustrate the synthetic data method but ultimately decided against it due to space constraints and the lack of added clarity.
>
> Additionally, Reviewer XMnz explicitly stated that “A more comprehensive set of practical demonstrations would significantly strengthen the argument for the proposed methods' utility.” We agree that adding relevant figures to address this concern is very reasonable. As such, we described the figures in our response, and we plan to add them once the revision is open.
>
> **In the end, the reviewer raised concerns regarding the rigor of the data distribution and difficulty level analysis.**
>
> Several new questions were introduced, such as "How were the difficulty levels of the data quantitatively defined and validated?" We believe this has already been clearly described in the manuscript, specifically in lines 439-445.
>
> The reviewer also brought up the issue of computational costs, requesting that we address the trade-offs in our design of comparative experiments to validate and discuss RQ3, which primarily focuses on exploring the differences in model capabilities. However, we feel that this request for a discussion of trade-offs is somewhat misplaced, as the focus of the experiment was not on trade-offs but on the differences in model abilities.
>
> Finally, we suggest referring to the more structured and coherent concerns raised by Reviewer fBsg regarding data distribution, along with our responses, as they provide a clearer foundation for addressing these issues.

---

> > ### Author Response · Authors · 2024-12-04
> > **Response to questions after clarifying concerns in Response 2 (1/2)**
> >
> > Next, we will respond to Response 2. We recommend first reviewing the earlier summary in "The Main Objection in Response 3 is Unreasonable," as most of the points raised here have already been clarified in that response.
> >
> > **The reviewer initially requested a further discussion of the theoretical justifications for the results.**
> >
> > In response to the original question "Q12: I recommend adding some theoretical backing or intuition as to why these methods work," we provided intuition in A12. Specifically, for the reviewer’s particular focus on the effectiveness of Tutorship Amplification, the intuition is as follows:
> > *“The effectiveness of Tutorship Amplification can be attributed to its ability to provide synthetic data that imparts self-correction capabilities to the model. By leveraging a teacher model to identify and correct errors in the student model’s responses, the data generated by this method endows the model with the ability to detect and correct errors, thereby enhancing its reasoning ability.”*
> >
> > We are unable to provide a theoretical justification for this intuition at this stage, as it may require starting with studying how LLM learns reasoning skills from data. In Section 5, we attempted a similar exploration starting from the conclusions. In Section 5.1, we reached Result 3: "Overall, while SFT can facilitate some learning of mathematical capabilities, it has a clear disadvantage compared to CPT." From here, we aimed to explore the factors contributing to these differences. Sections 5.2 and 5.3 delve into the dimensions of data distributions and difficulty levels, concluding that providing hard multi-step problem-solving data enables more effective learning, with this advantage being particularly evident in CPT compared to SFT.
> >
> > However, we also acknowledge that this conclusion is not final. We explicitly state, *"Regarding Result 5, why hard training data results in learning capability differences between CPT and SFT is not well-understood and may require theoretical analysis, demanding meticulous work. Similar work from a theoretical perspective seems to have just begun (Understanding Chair of That in LLMs Through Information Theory)."*
> >
> > **The reviewer next requested details regarding the specific thresholds or criteria used for deduplication and decontamination.**
> >
> > The details for deduplication and decontamination were already provided in Appendix A: Detailed Experimental Preparation, where we stated:
> > *"Specifically, we set a threshold of 2048 bytes for deduplication to improve the quality of the training data. Additionally, we set a threshold of 100 bytes to remove any data from the training set that contains more than 100 bytes of overlapping text with subsets of the train and test sets in the evaluation data. We believe this can account for some contamination caused by simple paraphrasing. (Notably, in the case of Openwebmath, we removed 2594 contaminated documents, which had a significant impact on the conclusions during our initial experiments.)"*
> >
> > The reviewer then seemed to shift focus to several unrelated issues, such as benchmarks, synthetic data, and reasoning steps evaluation. We recommend referring to the corresponding sections for these topics: the description of benchmarks can be found in Section 2: Experimental Preparation, the synthetic data discussion is in Section 4, and the description of reasoning steps is detailed in Section 5.3.

---

> > > ### Author Response · Authors · 2024-12-04
> > > **Response to questions after clarifying concerns in Response 2 (2/2)**
> > >
> > > **The reviewer next suggested that more discussion is needed regarding the real-world applicability of JiuZhang-8B.**
> > >
> > > In our previous responses, we summarized that JiuZhang-8B serves three primary purposes:
> > > 1. To validate the research conclusions presented in the paper.
> > > 2. To act as a foundational model that introduces a paradigm shift in training math-specific models, with features worth further exploration.
> > > 3. To demonstrate competitive performance as a math-specific base model within 8B parameters.
> > >
> > > Regarding the first point, the reviewer criticized it as circular reasoning. However, as we mentioned in lines 480-490 of the manuscript, we validated our conclusions by training on the current state-of-the-art model, LLaMa3-8B, and achieved good results. This serves as a validation, demonstrating that our conclusions are not confined to previously "weak" base models.
> > >
> > > For the second point, the reviewer asked for further clarification on the term "paradigm shift." As we discussed earlier, *"A paradigm shift in training math-specific models refers to acquiring more problem-solving data and moving it from the SFT phase to the CPT phase, directly targeting the learning of reasoning skills. This shift replaces the current practice of focusing the CPT phase on collecting internet-scale data and filtering large amounts of mathematical corpora for learning math knowledge. This change leads to a significant improvement in efficiency. Furthermore, combining with synthetic data methods, it can be generalized into an iterative data-model update process, where model capabilities and data quality alternately improve."*
> > >
> > > The reviewer concluded by stating that "A meaningful paper should stand on its own by addressing potential confounding factors."
> > >
> > > In response, we would like to emphasize that our study on the three research questions follows strict comparative experiments, where the training settings adhere to what are currently considered reasonable empirical setups. We have implemented rigorous variable control and reported the parameters used in the experiments. Furthermore, we provided additional details about the design of the comparative experiments in the Appendix: Discussion on Experimental Setting.
> > >
> > > If the reviewer’s criticism pertains to any potential confounding factors, we believe it should be more specific. We encourage the reviewer to pinpoint which particular research question (RQ) and which set of experiments they believe may have unreasonable settings, so that we can address those concerns more directly and constructively.
> > >
> > >
> > > **The reviewer subsequently pointed out that "A meaningful paper should stand on its own by addressing potential confounding factors."**
> > >
> > > We would like to emphasize that our study on the three research questions follows strict comparative experiments. The training setups are based on current reasonable empirical settings, with rigorous variable control, and we have reported the parameter configurations used. Furthermore, we provided additional details on the experimental design rationale in the Appendix: Discussion on Experimental Setting.
> > > Given this, the reviewer's criticism from this angle should be more specific. It would be helpful to clearly identify which research question (RQ) and which experimental setup are being referred to, and specify what potential confounding factors or unreasonable settings might be present.

---

### Official Review · Reviewer_fBsg · 2024-11-09

**Soundness:** 3
**Presentation:** 4
**Contribution:** 3
**Rating:** 8
**Confidence:** 4

**Summary:**

This paper investigates the impact of incorporating problem-solving data, various data synthesis techniques, and different training stages on enhancing mathematical reasoning capabilities in large language models (LLMs). The authors examine whether problem-solving data improves continued pre-training (CPT) effectiveness over traditional mathematical corpora and explore the optimal mixture ratios of such data. The study also assesses four data synthesis methods—response diversification, query expansion, retrospective enhancement, and tutorship amplification—highlighting the latter as particularly effective. Additionally, the authors compare the mathematical skills acquired during CPT versus supervised fine-tuning (SFT), finding that CPT leads to stronger mathematical reasoning capabilities, especially on complex, multi-step problems. These insights are applied to create JiuZhang-8B, a model that achieves competitive performance on mathematical reasoning benchmarks against other math-specific models.

**Strengths:**

1. The paper provides an in-depth analysis of the effects of problem-solving data on mathematical reasoning, particularly through comparisons of CPT and SFT, with valuable insights into the role of SFT’s instruction-following capabilities.

2. The authors explore and  compare different data synthesis methods, which is useful. The insights shared on the performance difference for the four different methods would be helpful for the LLM Reasoning community.

3. The comparison between models trained on SFT and CPT at varying difficulty levels adds further depth to understanding model capabilities.

4. Most of the sections are clearly written and well-structured, with detailed methodology on dataset curation and training, making it accessible and informative for the research community.

5. The final results of JiuZhang-8’s performance compared to SOTA LLMs is impressive, validating the insights and techniques shared earlier. The release of the model’s base version would greatly support community development.

**Weaknesses:**

1. The section on exploring the impact of different data distributions (5.2) is confusing and needs more clarification:
    1. While the authors note the inherent overlap in mathematical knowledge and the challenge in aligning data distribution with specific capabilities, the extent of overlap between knowledge points remains unclear. A dataset similarity analysis could be helpful to address questions raised below.
    2. The capability analysis in Figure 3 suggests that GAOKAO’s knowledge capabilities  subsumes MATH, while ZHONGKAO encompasses questions requiring higher general knowledge capabilities than MATH. This should mean that a model trained to perform better on GAOKAO should perform even better in MATH than a model trained to perform better in ZHONGKAO. However this is not the case, The authors designate middle school data as IND to ZHONGKAO and high school data as IND to GAOKAO, yet models trained on middle school data (both SFT and CPT) outperform high school-trained models on the MATH evaluation set (see Table 2). This raises questions about: (a) how the authors define OOD capabilities—whether based on the absence of similar problems in training or lack of shared underlying concepts; and (b) the effectiveness of the knowledge point labels used for segmenting middle and high school data.
   3. Figure 3 indicates that ZHONGKAO includes more advanced (level 3) general knowledge questions than GAOKAO. Since both NuminaMath and Lila contains math problem-solving data, assuming the proprietary dataset also contains mostly Math problem solving data,  a more detailed analysis of why high school math-trained models perform significantly worse on the ZHONGKAO evaluation would be beneficial.
   4. More granular difficulty-level analysis of the middle school and high school datasets, as applied in Section 5.3, would enhance clarity.


2. Result 8 appears somewhat trivial and lacks novelty, given that prior research has shown LLMs—and even simpler neural networks—tend to learn simpler representations first.

3. Results 4 and 6 appear redundant. Result 4 (line 377) highlights that SFT is less effective than CPT in learning mathematical skills, while Result 6 (lines 427-428) conveys a similar conclusion, stating that SFT's in-domain learning ability is weaker than CPT’s.

4. It seems to be that the majority of the problem solving data seems to be proprietary, thus making it hard to reproduce.

**Questions:**

See weakness for questions

1. The authors claims of collecting 25 million problem solving data. Assuming authors collected approximately 930,000 pieces from NuminaMath [1] and 140,000 from Lila [2], Does it mean that the proprietary problem solving data alone contains ~24 million pieces? or are the pieces calculated differently from the dataset websites?

2. A brief summary on how Figure 3 (Ability dimensions of four evals) is computed would be helpful in understanding Section 5.2.

[1] https://huggingface.co/collections/AI-MO/numinamath-6697df380293bcfdbc1d978c

[2] https://lila.apps.allenai.org

---

> ### Author Response · Authors · 2024-11-18
> **Initial Response to Reviewer Comments (1)**
>
> Thank you for your review! We greatly appreciate your valuable feedback and the issues you have pointed out. First, we will provide an overall discussion on the experimental approach for RQ3 and clarify the three concepts that may have caused confusion. Subsequently, we will address the specific issues raised. In addition, we have reuploaded the revised manuscript and included supplementary content in Appendix B.
>
> ## Clarification of the Experimental Approach and Related Results for RQ3
> Thank you for your suggestions on the expression of RQ3. We originally recorded the results according to **the following experimental logic** and hope for clarification.
> Finally, based on the suggestions, we have **revised the conclusion in the PDF** to make it clearer.
>
> For RQ3, we first explored the overall differences in capabilities developed during the CPT and SFT stages in **Section 5.1**. This led to **Result 4**, indicating that SFT is less effective than CPT in learning mathematical skills.
> In the following two subsections, we aimed to investigate the sources of this difference from various perspectives:
> - **Section 5.2**: We hypothesized that CPT might contribute to enhanced out-of-distribution (OOD) performance. However, comparative experiments revealed that both SFT and CPT primarily develop capabilities aligned with their respective data distributions, but SFT’s in-domain (IND) learning ability is weaker than CPT's (**Result 6**).
> - **Section 5.3**: Since **Result 6** was more pronounced in high school training data compared to middle school, this inspired us to examine how differences in training data difficulty contribute to the learning disparities between SFT and CPT. We re-segmented the dataset based on the number of steps required to solve problems. This experimentation yielded **Result 7**, showing that the difference primarily arises because CPT is better at learning from hard data, which partially addresses RQ3. **Result 8**, in contrast to findings based on data distribution, focused on difficulty dimensions. Unlike the data distribution perspective, where providing corresponding data improves IND capabilities, in the difficulty dimension, even when hard data is provided, it does not directly enhance the ability to solve hard problems. Instead, the model primarily focuses on learning to solve simpler, fewer-step problems.
>
>
> ## Three Concepts: Capability Dimensions, Data Distribution, and Difficulty Level
>
> Thank you for pointing out some issues with these three concepts, which seem indeed to have caused confusion. We need to clarify that these were roughly abstracted based on necessity during the research process and were not derived from any existing standards. The absence of such standards necessitates further discussion and harmonization by future work.
>
> ### Capability Dimensions
> Concerning the capability dimensions of the evaluation set, we introduce these dimensions before the specific research question to define three aspects: **general knowledge**, **math knowledge**, and **reasoning steps**. These dimensions help to understand the model results beyond just the average accuracy.
>
> ### Data Distribution
> Regarding the concept of data distribution, in Section 5.2, we explore the capability training differences between **SFT** and **CPT** for in-domain and out-of-distribution data by selecting and splitting the training data to represent the data distribution of the evaluation set. This concept is generally considered to be strongly related to the evaluation dimension of **math knowledge**, but in reality, only when both questions belong to a very fine-grained math knowledge can they be considered to have the same data distribution. As you mentioned, introducing similarity directly from the data distribution itself might be a better approach.
>
> Due to the large size of the training data, it is difficult to apply similarity analysis and ensure the accuracy of numerous fine-grained labels. Hence, we have opted for a **high-level** label, hoping to cover the data distribution of the evaluation set to explore in-domain capabilities.
>
> ### Difficulty Level
> The concept of difficulty is relatively clear. In Section 5.3, we re-segment the training data subsets using the **number of reasoning steps** in problem-solving data to explore the differences in training data of varying difficulties at different training stages.
>
> ### Supplementary Information
> We have supplemented **Appendix B** of the revised paper with:
> - **Examples from the evaluation set**,
> - **Simple definitions of the three dimensions**,
> - **Data distribution of the evaluation set problems**, and
> - **The similarity calculated based on this distribution**.
>
> However, due to the advantage of the training data in covering a broad range of diverse data from the evaluation set, it is quite unstable to analyze through a small number of samples. We hope the added information and discussions clarify the content.

---

> > ### Author Response · Authors · 2024-11-18
> > **Initial Response to Reviewer Comments (2)**
> >
> > ### Response to Specific Questions
> >
> > **Q1**:  (a) How do the authors define OOD capabilities—whether based on the absence of similar problems in training or the lack of shared underlying concepts; and  (b) the effectiveness of the knowledge point labels used for segmenting middle and high school data.
> >
> > **A1**:  The definitions of data distributions and in-domain capabilities remain somewhat ambiguous within the community. We believe this should at least encompass the finest-grained knowledge points, such as "functions-linear functions-solving linear functions." At the same time, problem knowledge point labels can be multiple and are challenging to ensure consistent quality. To address this issue, we adopted a straightforward approach: categorizing the data based on the source of the problems into "unknown," "primary school," "middle school," and "high school" data.
> >
> > Regarding the effectiveness of this categorization, due to the advantage of a sufficiently large training dataset providing broad coverage of knowledge points, it is difficult to evaluate similarity through sampling as in a test dataset. Nevertheless, from the IND results corresponding to Section 5.2, the original labels demonstrate a certain degree of effectiveness. Moving forward, training a classification model to validate or annotate knowledge point labels and grade-level labels may be a viable option for further improvement.
> >
> > **Q2**:  A model trained to perform better on GAOKAO should perform even better in MATH than a model trained to perform better in ZHONGKAO.
> >
> > **A2**:  This is indeed a peculiar phenomenon. Beyond the dimension of capability, the correlation analysis of the evaluation datasets (newly added to Appendix B) indicates that GAOKAO is also more correlated with MATH than ZHONGKAO, albeit to a limited extent. This phenomenon, similar to the differences in other OOD capabilities triggered by middle school data and high school data discussed in Section 5.2, may result from the combined influence of multiple dimensions of capability.  This includes the next issue mentioned—high school math-trained models performing significantly worse on the ZHONGKAO evaluation—as well as their varying performances on GSM8K. Section 5.2 provides some consistent conclusions regarding IND capabilities, but the comparative experimental design it encompasses still makes it difficult to pinpoint the source of these differences. Further work is needed to more clearly discuss and investigate the impact of data distribution on various capabilities, especially OOD capabilities.
> >
> > **Q3**:  Figure 3 indicates that ZHONGKAO includes more advanced (level 3) general knowledge questions than GAOKAO. Since both NuminaMath and Lila contain math problem-solving data, assuming the proprietary dataset also contains mostly math problem-solving data, a more detailed analysis of why high school math-trained models perform significantly worse on the ZHONGKAO evaluation would be beneficial.
> >
> > **A3**:  This issue shares a similar background with the previous question. Although GAOKAO is generally considered to encompass the mathematical knowledge of ZHONGKAO, this does not imply that their problem distributions are identical, as the data distribution similarity is approximately 0.33. Other capability dimensions, including the mentioned general knowledge, also influence performance.
> >
> > Comparing high school math-trained models and middle school math-trained models on the ZHONGKAO evaluation may not be entirely appropriate, as the latter's advantage likely stems from providing IND data. The impact of general knowledge is more apparent in the GSM8K evaluation. Although GSM8K is negatively correlated with ZHONGKAO and GAOKAO in terms of dataset distribution, the differences are limited, and no corresponding training data has been provided.
> >
> > From a capability perspective, the mathematical knowledge required for GSM8K primarily involves basic arithmetic, which has minimal impact. However, middle school math-trained models perform better on GSM8K. From the remaining dimensions, this could be attributed to middle school data encompassing more general knowledge.
> >
> > The following is a typical case of a problem caused by a lack of understanding of wages:
> > - **GSM8K Question**:
> >   *Eliza's rate per hour for the first 40 hours she works each week is $10. She also receives an overtime pay of 1.2 times her regular hourly rate. If Eliza worked for 45 hours this week, how much are her earnings for this week?*
> >
> > - **Middle School Math-Trained Model's Response**:
> >   *Eliza's earnings for this week are calculated as follows: 40 hours at $10 per hour plus 5 hours at $12 per hour. The total earnings are $400 + $60 = $460.*
> >
> > - **High School Math-Trained Model's Response**:
> >   *Eliza's regular hourly rate is $10. For overtime, she earns 1.2 times her regular rate. So for 45 hours, she earns 45 * 1.2 = 54 dollars. The answer is 54.*

---

> > > ### Author Response · Authors · 2024-11-18
> > > **Initial Response to Reviewer Comments (3)**
> > >
> > > **Q4**:  More granular difficulty-level analysis of the middle school and high school datasets, as applied in Section 5.3, would enhance clarity.
> > >
> > > **A4**:  This question appears to involve some misunderstanding. Section 5.3 re-segments the dataset based on reasoning steps to explore the impact of difficulty, as described in Lines 439-445:
> > > *"We selected a 5B subset of our problem-solving data and categorized it based on the number of solution reasoning steps: data requiring 1-3 steps was classified as easy, 4-7 steps as medium, and 8 or more steps as hard. The distribution of samples accounted for 36.0%, 38.4%, and 25.6% of the total data, respectively, while token counts made up 23.0%, 36.0%, and 41.0%, respectively."*
> > >
> > > In contrast, Section 5.2 segments the data into middle school and high school subsets based on data distribution. The average step counts were 5.74 for the middle school subset and 6.85 for the high school subset, respectively. Although these subsets are somewhat related to the inference steps, they are not the basis for segmentation in Section 5.3.
> > >
> > > **Q5**:  How is the quantity of problem-solving data calculated?
> > >
> > > **A5**:  The method for calculating the quantity of data is not particularly unique. We indeed have over 24 million pieces of proprietary data. Compared to the available open-source problem-solving datasets, this scale might seem exaggerated. However, it corresponds to only 7.2 billion tokens, which, when compared to the current scale of pretraining datasets, still leaves room for growth. We plan to continue expanding this dataset. During the development of JiuZhang-8B, we attempted to expand the dataset to 35 million pieces, but the improvement was limited. This may indicate that we have reached a stage where quality outweighs quantity or that we have encountered a capability ceiling imposed by the current dataset. This is a topic worth further investigation.
> > >
> > > For the datasets from NuminaMath and Lila, there is a slight discrepancy in the usage scale, which we clarify here. NuminaMath showed evident test set contamination, with approximately 40,000 samples removed. After accounting for duplicates, the final dataset used consisted of 858,000 samples. For Lila, the data was not based on queries but rather on the provided query-response pairs, resulting in 317,000 samples.

---

### Meta-Review · Area_Chair_wUMN · 2024-12-24

**Metareview:**

This paper took an ambitious goal of identifying the impact of data, methods, and training stages in advancing the math capability of LLMs.  The author explores alternative strategies during the pre-training phase, focusing on the use of problem-solving data and examines whether they improve continued pre-training (CPT) effectiveness over traditional mathematical corpora and explore the optimal mixture ratios of such data. The study also assesses four data synthesis methods, highlighting the latter as particularly effective. Additionally, the authors compare the mathematical skills acquired during CPT versus supervised fine-tuning (SFT), finding that CPT leads to stronger mathematical reasoning capabilities, especially on complex, multi-step problems.

This paper has received rather polarizing reviews. After extensive discussion with the reviewers and careful consideration, we still recommend acceptance given its merits in the extensiveness of study.

**Additional Comments On Reviewer Discussion:**

This paper received 7 official reviews. 2 of them were negative while 5 were rather positive. One negative reviewer (who rated 3) never engaged during the discussion phase, and hence I discounted that review slightly. 2 reviewers have updated (raised) their scores durign rebuttal.

---

### Decision · Program_Chairs · 2025-01-22

Accept (Poster)